

# Chemistry-climate feedback of atmospheric methane in a methane emission flux driven chemistry-climate model

Laura Stecher[1], Franziska Winterstein[1], Patrick Jöckel[1], Michael Ponater[1], Mariano Mertens[1], and Martin Dameris[1]

[1]Deutsches Zentrum für Luft- und Raumfahrt, Institut für Physik der Atmosphäre, Oberpfaffenhofen, Germany

**Correspondence:** Laura Stecher (laura.stecher@dlr.de)

**Abstract.**

The chemical sink of atmospheric methane ($CH_4$) depends on the temperature and on the chemical composition. Here, we assess the feedback of atmospheric $CH_4$ induced by changes of the chemical sink in a warming climate using a $CH_4$ emission flux driven setup of the chemistry-climate model EMAC, in which the chemical feedback of $CH_4$ mixing ratios can evolve explicitly. We perform idealized perturbation simulations driven either by increased carbon dioxide ($CO_2$) mixing ratios, or by increased $CH_4$ emission fluxes. The $CH_4$ emission flux perturbation leads to a large increase of $CH_4$ mixing ratios. Remarkably, the factor by which the $CH_4$ mixing ratio increases is larger than the increase factor of the emission flux, because the atmospheric lifetime of $CH_4$ is extended.

In contrast, the individual effect of the global surface air temperature (GSAT) increase is to shorten the $CH_4$ lifetime, which results in a significant reduction of $CH_4$ mixing ratios in our setup. The corresponding radiative feedback is estimated at -0.041 W m$^{-2}$ K$^{-1}$ and -0.089 W m$^{-2}$ K$^{-1}$ for the $CO_2$ and $CH_4$ perturbation, respectively. The explicit adaption of $CH_4$ mixing ratios leads to secondary feedbacks of the hydroxyl radical (OH) and ozone ($O_3$). Firstly, the OH response includes the $CH_4$-OH feedback, which enhances the $CH_4$ lifetime change, and, secondly, the formation of tropospheric $O_3$ is reduced. Our $CH_4$ perturbation induces the same response of GSAT per effective radiative forcing (ERF) as the $CO_2$ perturbation, which supports the applicability of the ERF framework for $CH_4$.

## 1 Introduction

Methane ($CH_4$) is, after carbon dioxide ($CO_2$), the second most important anthropogenic greenhouse gas (GHG). Compared to $CO_2$, $CH_4$ has a larger radiative efficiency (Forster et al., 2021) and a shorter atmospheric lifetime of about 10 years (e.g., Prather et al., 2012; Stevenson et al., 2020). Therefore, reducing atmospheric $CH_4$ mixing ratios is considered an important measure to mitigate climate change on a decadal time scale (Saunois et al., 2016; Collins et al., 2018; Ocko et al., 2021; Staniaszek et al., 2022). The relative short atmospheric lifetime of $CH_4$ is a consequence of the fact that $CH_4$ is a chemically active species. According to Saunois et al. (2020), the most important sink of atmospheric $CH_4$ is the oxidation with the hydroxyl radical (OH). Thus, understanding the chemical mechanisms underlying the $CH_4$ oxidation is crucial when assessing its climate impact and mitigation options.



Besides its direct radiative effect, indirect contributions of ozone ($O_3$) and stratospheric water vapour ($H_2O$) enhance the effective radiative forcing (ERF) of $CH_4$ (Shindell et al., 2005, 2009; Stevenson et al., 2013; Winterstein et al., 2019; Thornhill et al., 2021b; O'Connor et al., 2022). In addition to its climate impact, tropospheric $O_3$ poses harmful effects on human health (Nuvolone et al., 2018) and on vegetation (Ashmore, 2005). Therefore, mitigation options involving $CH_4$ emission reduction have beneficial effects on air quality (Shindell et al., 2012; Staniaszek et al., 2022) and plant productivity (Sitch et al., 2007). In

addition to the effects on $O_3$ and $H_2O$, the $CH_4$ oxidation reduces OH, which feeds back onto its own atmospheric lifetime (e.g., Winterstein et al., 2019) and affects the rate of formation of secondary aerosols leading to a shift in the aerosol-size distribution. The latter, in turn, influences aerosol-radiation interactions and aerosol-cloud interactions, and is considered another indirect contribution to the ERF of $CH_4$ (Kurtén et al., 2011; O'Connor et al., 2022).

Next to its importance for indirect contributions to the ERF, the $CH_4$ oxidation largely constrains the atmospheric lifetime of

$CH_4$ and, thus, together with the magnitude of the emissions, its direct radiative effect. The atmospheric lifetime of $CH_4$ is not constant, but depends on the temperature and on the chemical background, which determines the abundance of its sink reactants, especially OH. OH is influenced by a multitude of factors (e.g., Voulgarakis et al., 2013; Stevenson et al., 2020). Among others, meteorological factors such as humidity and temperature influence the abundance of OH. Hence, climate feedbacks of the chemical sink of $CH_4$ and thereby its lifetime are to be expected. More precisely, the $CH_4$ lifetime is projected to shorten as

a result of tropospheric warming (Voulgarakis et al., 2013; Stecher et al., 2021; Heimann et al., 2020; Thornhill et al., 2021a). Up to date, only a limited number of studies have assessed the corresponding response of $CH_4$ mixing ratios directly (Heimann et al., 2020). Even though efforts are ongoing to employ chemistry-climate models (CCMs) in a $CH_4$ emission flux driven model setup (Shindell et al., 2013; He et al., 2020; Folberth et al., 2022), it is still common practice to prescribe $CH_4$ mixing ratios at the lower boundary. For instance, the latter method was pursued in the Aerosol Chemistry Model Intercomparison

Project (AerChemMIP; Collins et al., 2017), which is endorsed in the Coupled-Model Intercomparison Project 6 (CMIP6; Eyring et al., 2016), which built the basis for the last IPCC report (IPCC, 2021).

When $CH_4$ mixing ratios are prescribed, the adjustment of $CH_4$ mixing ratios to changes of the chemical sink is suppressed, and can be only derived offline from the atmospheric lifetime change (Dietmüller et al., 2014; Heinze et al., 2019; Thornhill et al., 2021a). This offline method, however, suppresses indirect feedbacks induced by the $CH_4$ response. On the one hand, the

resulting $CH_4$ response in turn alters the atmospheric $CH_4$ lifetime, which leads to subsequent adaptions of the $CH_4$ mixing ratios. The derivation of the $CH_4$ response from the lifetime change usually accounts for this effect by including a constant $CH_4$-OH feedback factor $f$ (Heinze et al., 2019; Thornhill et al., 2021a) so that the equilibrium $CH_4$ mixing ratio $[CH_4]_{eq}$ is estimated as (e.g., Stevenson et al., 2020)

$$[CH_4]_{eq} = [CH_4]_{ref}(\frac{\tau_{exp}}{\tau_{ref}})^f, \tag{1}$$

where $[CH_4]_{ref}$ is the reference $CH_4$ mixing ratio and $\tau_{ref}$ and $\tau_{exp}$ are the reference and perturbed atmospheric lifetimes of $CH_4$, respectively. Estimates of $f$ are in the range of 1.2 to 1.4 (Fiore et al., 2009; Voulgarakis et al., 2013; Stevenson et al., 2013; Thornhill et al., 2021b; Stevenson et al., 2020). Holmes (2018) found that $f$ can vary geographically and seasonally, and that it strengthens with an increasing $CH_4$ burden. On the other hand, the subsequent $CH_4$ response affects other chemical



constituents such as $O_3$. This effect is also sometimes accounted for by scaling the sensitivity of $O_3$ towards $CH_4$ perturbations
with the expected $CH_4$ response (Fiore et al., 2009; Thornhill et al., 2021b). However, previous studies that assessed the
climate feedback of $O_3$ and the corresponding implication for the global surface air temperature (GSAT) response did not, or
only rudimentarily, account for the interaction between changes of $O_3$ and $CH_4$ (Dietmüller et al., 2014; Nowack et al., 2015;
Marsh et al., 2016; Li and Newman, 2023). As for $O_3$ alone the latter studies indicate that its feedback reduces the resulting
GSAT response, or in other words that it is a negative feedback. The magnitude of this feedback, however, has been highly
model dependent.

In this study, we use a $CH_4$ emission flux driven setup of the CCM ECHAM/MESSy Atmospheric Chemistry (EMAC; Jöckel
et al., 2016) to explicitly simulate the response of atmospheric $CH_4$ mixing ratios resulting from changes in the chemical sink.
We perform idealized perturbation simulations with either increased $CO_2$ mixing ratios (with $CO_2$ being an inert GHG), or
increased $CH_4$ emission fluxes. The $CH_4$ perturbation affects the chemical composition directly, whereas the $CO_2$ perturbation
influences the chemical composition only indirectly, e.g. through the temperature change. From the simulation results we assess
the change of $CH_4$ mixing ratios and its implications for OH and $O_3$. Next to the $CH_4$ feedback, GSAT changes induce other
processes that influence $O_3$. For instance, temperature changes affect chemical reaction rates, emissions of precursor species
from natural sources, circulation, or the abundance of $H_2O$ (e.g., Stevenson et al., 2006; Chiodo et al., 2018; Abalos et al.,
2020; Griffiths et al., 2021; Zanis et al., 2022). Therefore, we apply an attribution method for $O_3$ (TAGGING; Grewe et al.,
2017; Rieger et al., 2018) to identify and quantify the importance of individual source categories that influence tropospheric
$O_3$ under tropospheric warming.

Our analysis is based on the ERF conceptual framework (Shine et al., 2003; Hansen et al., 2005; Ramaswamy et al., 2018;
Forster et al., 2021), which means that the so-called fast and (slow) climate responses are assessed separately. The fast response
represents the part of the full response that develops, on short time scales, independent of the corresponding GSAT change,
whereas the climate response represents the isolated effect of the GSAT change. There is no formal time scale that separates the
fast and slow responses, but they are distinguished conceptually by the dependence on the GSAT response, which is coupled
to the (slow) response of the ocean. It is noteworthy that the $CH_4$ adjustment that follows the increase of $CH_4$ emission fluxes
evolves on the time scale of decades, whereas typical physical (rapid) adjustments evolve on the time scale of weeks or months
(e.g., Smith et al., 2018). We derive ERF and the fast response from simulations with prescribed sea surface temperatures (SSTs)
and sea ice concentrations (SICs) as recommended by Forster et al. (2016). The climate response is assessed as the difference
between the response in a simulation coupled to a mixed layer ocean model and the respective fast response. Analogously, we
assess (rapid radiative) adjustments as the radiative effects corresponding to changes in the fast response and (slow climate)
feedbacks as the radiative effects corresponding to changes in the climate response.

The paper is structured as follows. Section 2 explains the simulation set-up (Sect. 2.1.), as well as the TAGGING method to
attribute $O_3$ to individual source categories (Sect. 2.2.), and the method to derive the radiative effects corresponding to com-
position changes of individual species (Sect. 2.3.). Additionally, Sect. 2.3. introduces the theoretical framework for radiative
forcing and climate sensitivity. Section 3 presents the simulation results. In Sect. 3.1. and 3.2., we present composition changes



in the $CO_2$ and $CH_4$ perturbation simulations, respectively. In Sect. 3.3., the corresponding radiative effects are addressed. We conclude with a general discussion and summary of our findings in Sect. 4.

## 2 Methods

### 2.1 Model description and simulation strategy

We use the modular CCM ECHAM/MESSy Atmospheric Chemistry (EMAC; Jöckel et al., 2016) in version 2.55.2. All simulations are performed at a resolution of T42L90MA, i.e. at a triangular (T) truncation at wave number 42 of the spectral dynamical core, corresponding to a quadratic Gaussian grid of approximately $2.8° \times 2.8°$ resolution in latitude and longitude and 90 vertical levels, with the uppermost level centered around 0.01 hPa. The simulations are performed as time slices, which means that the same boundary conditions are repeated cyclically for each simulation year. The boundary conditions, i.e. prescribed emissions and mixing ratios of well-mixed GHGs, represent the year 2010. The quasi-biennial oscillation (QBO) is nudged following the method of Giorgetta and Bengtsson (1999) as described by Jöckel et al. (2016), which introduces some interannual variability. The simulation set-up builds on previous studies assessing the impact of enhanced $CH_4$ mixing ratios with EMAC (Winterstein et al., 2019; Stecher et al., 2021), with the important advance that for the present study EMAC is used in a $CH_4$ emission flux driven setup, which means that $CH_4$ emission fluxes instead of $CH_4$ mixing ratios are prescribed at the lower boundary. The $CH_4$ emission fluxes are prescribed as offline emission fluxes, i.e. there are no feedbacks on $CH_4$ emission fluxes from natural sources such as wetlands or permafrost, which are expected to change in a changing climate (e.g., O'Connor et al., 2010; Dean et al., 2018).

The applied $CH_4$ emission inventory is an inverse optimized inventory for the EMAC model (Frank, 2018). For the two reference simulations (see Tab. 1 and text below), monthly mean emissions of the year 2010 are repeated cyclically and scaled by a globally constant factor of 1.08, which corresponds to total annual mean emissions of 625.3 $Tg(CH_4)$ $a^{-1}$. The scaling was applied to bring the simulated $CH_4$ surface mixing ratios closer to observations. As the tropospheric mean $CH_4$ lifetime is about 10 years (e.g., Prather et al., 2012; Stevenson et al., 2020), the $CH_4$ mixing ratios of the year 2010 result not only from $CH_4$ emissions of the year 2010, but also from emissions of the years before. Therefore, it can not be expected that the cyclic repetition of $CH_4$ emissions of the year 2010 results in $CH_4$ mixing ratios that represent the year 2010 exactly. Applying the scaling, the resulting global mean $CH_4$ surface mixing ratio is 1.82 ppmv for both reference simulations. This is in close agreement with observational estimates for the years 2010 and 2012 of 1.80 and 1.81 ppmv by NOAA/ESRL (Lan et al., 2023), and 1.81 and 1.82 ppmv by the WMO World Data Centre for Greenhouse Gases (WMO, 2022). The estimates of NOAA/ESRL tend to be lower, as only unpolluted marine surface sites contribute to the global estimate. The chemical sink reactions of $CH_4$ with OH, excited oxygen ($O(^1D)$) and chlorine (Cl), and $CH_4$ photolysis are interactively accounted for by the MESSy submodels Module Efficiently Calculating the Chemistry of the Atmosphere (MECCA; Sander et al. (2019)) and JVAL (Sander et al., 2014). In addition to the sink reactions of $CH_4$, the chemical mechanism covers the basic chemistry of $O_3$, OH, hydroperoxyl ($HO_2$), nitrogen oxides, alkanes and alkenes up to four C-atoms, and isoprene ($C_5H_8$). Further, halogen chemistry of bromine and chlorine species is included. Alkynes, aromatics and mercury are not considered. In total,



the used mechanism covers 265 gas-phase, 82 photolysis and 12 heterogeneous reactions of 160 species. The soil sink of $CH_4$ is included by the submodel DDEP (Kerkweg et al., 2006a), which uses a prescribed deposition rate (Spahni et al., 2011; Curry, 2007) that is scaled to the actual $CH_4$ mixing ratio in the corresponding grid box. On average, the global soil sink is 27.6 $Tg(CH_4)$ $a^{-1}$ for both reference simulation.

Precursor emissions of $O_3$, in particular nitrogen oxides NO and $NO_2$ ($NO_x$), non-methane hydrocarbons (NMHCs), and carbon monoxide (CO), are treated as described by Jöckel et al. (2016). Anthropogenic emissions of these species are prescribed from the MACCity inventory (Lamarque et al., 2010; Granier et al., 2011; Diehl et al., 2012), whereby the (mostly monthly resolved) emission fluxes of the year 2010 are repeated cyclically. In addition, climatologies of biogenic emissions of NMHC and CO are prescribed from the Global Emissions InitiAtive (GEIA). Natural emissions of $NO_x$ from lightning, $NO_x$ and
$C_5H_8$ from biogenic sources, as well as the exchange of chemical species between atmosphere and ocean are parameterized. For lightning $NO_x$ the parameterization of Grewe et al. (2001) is used in the MESSy submodel LNOX (Tost et al., 2007). The 20-year mean global emissions from lightning $NO_x$ are 5.2 $Tg(N)$ $a^{-1}$ for both reference simulations (see Tab. 3). Interactive biogenic emissions of soil $NO_x$ and $C_5H_8$ are calculated by the submodel ONEMIS (Kerkweg et al., 2006b). On average, biogenic $NO_x$ emissions are 6 $Tg(N)$ $a^{-1}$ and biogenic $C_5H_8$ emissions are about 307 $Tg(C)$ $a^{-1}$ for both reference simulations
(see Tab. 3). The atmosphere-ocean exchange of the chemical species $C_5H_8$, dimethyl sulfide (DMS) and methanol ($CH_3OH$) is parameterized using the submodel AIRSEA (Pozzer et al., 2006).

Table 1 summarizes the performed simulations. The simulations REF-SSTfix and REF-SSTvar serve as references for the experiment simulations and represent year 2010 conditions. REF-SSTfix is performed using prescribed SSTs and SICs, whereas for REF-SSTvar a mixed layer ocean (MLO) model is coupled (MESSy submodel MLOCEAN, Kunze et al. (2014); original
ECHAM5 code by Roeckner et al. (1995), described in the ECHAM5 documentation (Chap. 6.3 – 6.5 in Roeckner et al., 2003)). The prescribed multi-year monthly mean climatology of SSTs and SIC is an observations based estimate of the years 2000 to 2009 from the Met Office Hadley Center (Rayner et al., 2003). The same climatology was used by Winterstein et al. (2019).

The perturbation simulations ERFCO$_2$ and ERFCH$_4$ are performed with the same prescribed climatology of SSTs and SICs
as REF-SSTfix to assess the so-called fast response and to quantify the ERF and the adjustments following the fixed SST method (e.g., Forster et al., 2016). ERF is defined as the top of the atmosphere (TOA) net radiative flux change between the experiment and the reference simulation. The perturbations of $CO_2$ and $CH_4$ are scaled to result in ERFs of similar magnitude, because for a most fair comparison of the climate sensitivity parameters of different perturbation agents, the respective forcings need to be at the same order of magnitude as the climate sensitivity, as it can be dependent on the magnitude of the forcing
(e.g., Hansen et al., 2005; Dietmüller et al., 2014). The ERF is targeted to be large enough to cause statistically significant and interpretable feedbacks (Forster et al., 2016), and small enough to be reached with still realistically large perturbations of $CO_2$ and $CH_4$ (Dietmüller et al., 2014; Winterstein et al., 2019). Perturbations of $1.35 \times CO_2$ mixing ratios and $2.75 \times CH_4$ surface emission fluxes result in ERFs of $1.61 \pm 0.16$ W m$^{-2}$ and $1.72 \pm 0.17$ W m$^{-2}$, respectively (see Tab. 4). The scaling of $1.35 \times CO_2$ mixing ratios or $2.75 \times CH_4$ surface emission fluxes is applied to all $CO_2$ and $CH_4$ perturbation experiments,
respectively. ECCCO$_2$ and ECCCH$_4$ are so-called equilibrium climate change simulations. In these simulations the MLO



model accounts for the response of SSTs and SICs. Therefore, the effect of GSAT driven feedbacks is included in these simulations.

In the following, we assess the so-called fast response as difference between ERFCO$_2$ or ERFCH$_4$ and REF-SSTfix, and the full response as difference between ECCCO$_2$ or ECCCH$_4$ and REF-SSTvar. The difference between the full and the fast

response is assessed as the climate response, which represents the isolated effect of the GSAT change. Similarly, adjustments and feedbacks are defined as the radiative effects corresponding to the fast response and the climate response, respectively. For the analysis, results of 20 simulated years after the spin-up period are used. The spin-up ensures that a quasi-equilibrium is reached and is, therefore, different for the individual simulations according to the simulation set-up. For ERFCH$_4$ the longest spin-up period of 90 years, for ECCCO$_2$ and ECCCH$_4$ 50 years, respectively, and for ERFCO$_2$ 25 years were necessary.

In this study, the CH$_4$ lifetime is calculated according to Jöckel et al. (2016) as

$$\tau_{CH_4} = \frac{\sum\limits_{b \in B} m_{CH_4}}{\sum\limits_{b \in B} k_{CH_4+OH}(T) \cdot c_{air}(T,p,q) \cdot x_{OH} \cdot m_{CH_4}}, \tag{2}$$

with $m_{CH_4}$ being the mass of CH$_4$ in [kg], $k_{CH_4+OH}(T)$ the temperature $T$ dependent reaction rate coefficient of the reaction CH$_4$ + OH → products in [cm$^3$ s$^{-1}$], $c_{air}$ the concentration of air in [cm$^{-3}$] and $x_{OH}$ the mole fraction of OH in [mol mol$^{-1}$] in all grid boxes b $\in$ B. B is the region, for which the lifetime should be calculated, e.g. all grid boxes below

the tropopause for the mean tropospheric lifetime. For the CH$_4$ lifetime calculation a climatological tropopause, defined as tp$_{\text{clim}}$= 300 hPa $-$ 215 hPa $\cdot$ cos$^2(\phi)$, with $\phi$ being the latitude in degrees north, is used as recommended by Lawrence et al. (2001). The reaction rate coefficient $k_{CH_4+OH}(T)$ is calculated as in the applied kinetic equation system (submodel MECCA), i.e. as

$$k_{CH_4+OH}(T) = 1.85 \times 10^{-20} \cdot T^{2.82} \cdot \exp\left(-\frac{987}{T}\right). \tag{3}$$

## 2.2 TAGGING

The TAGGING method (Grewe et al., 2017; Rieger et al., 2018) quantifies the contributions of individual source categories to the mixing ratios of tagged tracers. Tagged tracers are O$_3$, CO, reactive nitrogen compounds (NO$_y$), peroxyacyl nitrate (PAN), NMHCs, OH, and HO$_2$. For computational reasons NMHCs and NO$_y$ are considered with a family approach. For these species or families of species the individual contributions of emission categories or source processes are calculated. In this study O$_3$

production from the following categories is considered:

- through photolysis of molecular oxygen (O$_2$) in the stratosphere (*O$_3$ stratosphere*),

- from emissions of lightning NO$_x$ (*O$_3$ lightning*),

- from biogenic precursor emissions, mainly soil NO$_x$ and C$_5$H$_8$, (*O$_3$ biogenic*),

- from products of the CH$_4$ decomposition (*O$_3$ CH$_4$*),




**Table 1.** Overview of performed simulations. REF indicates that the respective reference is used, which is 388.4 ppmv for the global mean surface mixing ratio of $CO_2$, and 625.3 Tg($CH_4$) a$^{-1}$ for the $CH_4$ surface emissions. The prescribed multi-year monthly mean climatology of SSTs and SIC is based on an observations based estimate of the years 2000 to 2009 from the Met Office Hadley Center (Rayner et al., 2003).

| simulation name | SST + SIC | $CO_2$ VMR | $CH_4$ surface emissions |
|---|---|---|---|
| REF-SSTfix | prescribed | REF | REF |
| REF-SSTvar | MLO | REF | REF |
| ERFCO$_2$ | prescribed | REF×1.35 | REF |
| ECCCO$_2$ | MLO | REF×1.35 | REF |
| ERFCH$_4$ | prescribed | REF | REF×2.75 |
| ECCCH$_4$ | MLO | REF | REF×2.75 |

– from products of the nitrous oxide ($N_2O$) decomposition ($O_3$ *$N_2O$*),

    – from biomass burning precursor emissions ($O_3$ *biomass burning*)

    – and from anthropogenic precursor emissions ($O_3$ *anthropogenic*).

The categories are the same as defined by Grewe et al. (2017), except for the category $O_3$ *anthropogenic*, which, in our study, combines $O_3$ production from all anthropogenic emissions, i.e. of the sectors industry, road traffic, shipping and aviation.

The tagged tracers (i.e. the individual contributions) undergo the same processes as the corresponding total species. These are transport, emissions, dry and wet deposition, and chemical production and loss (see Grewe et al. (2017) for details). For the short-lived species OH and HO$_2$ a steady-state between chemical production and loss is assumed (Rieger et al., 2018). The chemical reaction rates are taken from the submodel MECCA. Effective production and loss is taken into account for $O_3$, meaning that production and loss terms from a family, which includes all fast exchanges between $O_3$ and other chemical
species, are considered. The diagnostic tool *ProdLoss* (Grewe et al., 2017) is used to identify all reactions that contribute to effective $O_3$ production and loss in the applied chemical mechanism. The reaction rates of effective $O_3$ production and loss are then manually grouped into $O_3$ production and loss rates, depending on which tagged species contributes to $O_3$ production or loss.

## 2.3 Quantification of individual radiative effects

The assessment of radiative effects in this study follows the ERF framework. This means that (rapid radiative) adjustments, which are defined as the TOA net radiative flux change corresponding to changes in the fast response, i.e. independent from GSAT changes, are accounted as part of the forcing. Thus, ERF is given as the sum of instantaneous radiative forcing (IRF), defined as the net radiative flux change at TOA excluding any adjustment, and the sum of all individual adjustments $A_i$ (e.g.,



Smith et al., 2018)

$$ERF = IRF + \sum_i A_i. \tag{4}$$

Consistently, (slow climate) feedbacks are defined as the TOA net radiative flux change induced by atmospheric parameter changes that correspond to the gradually changing GSAT. These feedbacks act to reduce or enhance the associated GSAT change ($\Delta T$) and determine the climate sensitivity parameter $\lambda$ (units: K / (W m$^{-2}$)), which is the proportionality constant that relates the equilibrium change of GSAT to the ERF as

$$\Delta T = \lambda \cdot ERF. \tag{5}$$

The feedback parameter $\alpha$ is the negative inverse of the climate sensitivity parameter $\lambda$ ($\alpha = -\frac{1}{\lambda}$; units: W m$^{-2}$ K$^{-1}$). The feedback parameter quantifies the net radiative flux change at TOA for a given change in GSAT. Under the assumption of linearity it can be decomposed into the radiative contributions of individual processes affected by the change in GSAT, i.e. the individual feedback parameters $\alpha_i$, so that (e.g., Forster et al., 2021)

$$\alpha = \sum_i \alpha_i = \sum_i \frac{\partial N}{\partial x_i} \frac{dx_i}{dT}, \tag{6}$$

where $N$ is the radiative flux change at TOA induced by the change of an individual variable of the Earth system $x_i$.

We quantify adjustments and feedbacks corresponding to composition changes of $CO_2$, $CH_4$, $O_3$ and stratospheric $H_2O$ following the method used by Winterstein et al. (2019) and Stecher et al. (2021): Additional simulations are performed with EMAC using the option for multiple diagnostic radiation calls (Dietmüller et al., 2016; Nützel et al., 2024). These simulations are performed (for the sake of saving computational resources) without interactive chemistry, but with prescribed climatologies for the radiatively active trace gases $CH_4$, $CO_2$, $O_3$, $N_2O$ and the chlorofluorocarbons (CFCs) from the simulations REF-SSTfix or REF-SSTvar. SSTs and SICs are prescribed using the same observational based climatology as used for REF-SSTfix (Rayner et al., 2003). Thus, the background climate of the simulations represents reference conditions. The additional simulations are run over 2 years each (plus 1 year spin-up).

In these simulations the first radiation call is used for providing the radiative heating rates that drive the base model, whereas the other radiation calls are purely diagnostic. The diagnostic radiation calls include the stratospheric temperature adjustment induced by the respective perturbation following the method of Stuber et al. (2001). The second, i.e. first diagnostic, call receives identical input as the prognostic radiation call, except for the specific humidity, for which a monthly mean climatology from the respective reference simulation is used instead of the prognostic specific humidity from the base model. The second call serves as the reference for the perturbations. It is necessary, because the subsequent diagnostic radiation calls, that are perturbed by monthly mean climatologies of the specific humidity from the perturbation simulations (see below), need to be compared to a radiation call, that uses a monthly mean climatology of the specific humidity from the reference simulation instead of the prognostic specific humidity. In addition, radiation calls are performed, for which either $CO_2$, $CH_4$, $O_3$ or the specific humidity are perturbed. The perturbed fields are monthly mean climatologies from the corresponding experiment simulations. For $O_3$ and the specific humidity the radiative effects of changes in the troposphere and in the stratosphere are derived





separately. The radiative effect corresponding to the perturbed $CH_4$ ($CO_2$) field from the simulation ERFCH$_4$ (ERFCO$_2$) repre­sents the respective estimate of stratospheric adjusted radiative forcing (SARF), i.e. the direct radiative effect of the perturbation including the associated stratospheric temperature adjustment.

There is one methodological difference compared to Winterstein et al. (2019) and Stecher et al. (2021). They used the
climatological specified humidity directly in the first prognostic radiation call, which then served as reference for the perturbed calls. However, here it was decided to use the prognostic specific humidity in the first radiation call as it is consistent with the model's background meteorology, e.g. the cloud cover. The influence on the calculated radiative effects was tested and found to be up to 1.02% (or 0.004 W m$^{-2}$) with the maximum deviation for the perturbations of specific humidity, which is negligible in comparison to other uncertainties.

## 3   Results

### 3.1   Methane and ozone composition changes following $1.35 \times CO_2$ perturbation

In this section, we present the simulation results of the $1.35 \times CO_2$ perturbation. Figure 1 shows the annual zonal mean com­position changes of $CH_4$ mixing ratios in the simulations ERFCO$_2$ (fast response) and ECCCO$_2$ (full response), and their difference, which is interpreted as the climate response. The fast response is dominated by increasing $CH_4$ mixing ratios in
the upper stratosphere and mesosphere. In this region, the cooling to be expected from $CO_2$ increase (see Fig. S1 in the sup­plement) leads to the prolongation of the $CH_4$ lifetime. A similar effect has been noted by Dietmüller et al. (2014). In the fast response, tropospheric $CH_4$ shows a slight increase below 2%.

In contrast to the fast response, the full response shows a significant reduction of $CH_4$ mixing ratios in the troposphere and lower stratosphere. As $CH_4$ emission fluxes are prescribed in the simulation set-up and cannot respond to changes in
meteorology or composition, any feedback of natural $CH_4$ emissions (e.g., Dean et al., 2018) is suppressed. Therefore, the decrease of $CH_4$ mixing ratios results from enhanced chemical decomposition of $CH_4$, mainly by the oxidation with OH. The tropospheric $CH_4$ lifetime with respect to the oxidation with OH shortens by about 7 months (0.56 a or 7.4 %, see Tab. 2). This shortening is a combined result of the direct influence of the temperature on the reaction rate coefficient and of an enhanced abundance of OH. Tropospheric warming increases OH mixing ratios throughout the troposphere with the maximum increase
in the tropics (see Fig. S3 in the supplement). The OH response is largely driven by the increase of tropospheric humidity associated with higher temperatures.

In addition to the reduction in the troposphere, the $CH_4$ mixing ratios decrease also in the lower stratosphere as part of the full response. As the reaction partners of $CH_4$, namely OH, Cl and O($^1$D), do not show any significant response in the lower stratosphere (see Fig. S3 for OH), the decrease is likely a transport effect. Tropospheric air masses with reduced $CH_4$ mixing
ratios compared to the reference simulation enter the stratosphere by the upwelling branch of the Brewer-Dobson-Circulation. Dietmüller et al. (2014) noted an increase of $CH_4$ mixing ratios throughout the stratosphere as a result of $2 \times CO_2$ due to stratospheric cooling and thereby a slower $CH_4$ oxidation. In addition, enhanced tropical upwelling transports $CH_4$ enriched





air in the stratosphere more efficiently. In our setup, the reduction of tropospheric $CH_4$, which is suppressed in the set-up of Dietmüller et al. (2014) as $CH_4$ mixing ratios are prescribed at the lower boundary, dominates the latter two processes.

Previous studies also found that tropospheric warming leads to increasing OH mixing ratios and correspondingly to the shortening of the $CH_4$ lifetime (e.g., Voulgarakis et al., 2013; Dietmüller et al., 2014; Heimann et al., 2020; Stecher et al., 2021; Thornhill et al., 2021a). The novelty of our study is that the associated reduction of $CH_4$ mixing ratios is simulated explicitly, which was assessed by only a small number of studies so far (Heimann et al., 2020). In our $CO_2$ perturbation experiment, the $CH_4$ lifetime change per unit change of GSAT is -0.51 a $K^{-1}$ or -6.7 % $K^{-1}$.

Voulgarakis et al. (2013) addressed the sensitivity of the $CH_4$ lifetime towards climate change in the Atmospheric Chemistry and Climate Modeling Intercomparison Project (ACCMIP) model ensemble. In the respective sensitivity simulations the boundary conditions for SSTs, SICs and $CO_2$ were set to RCP8.5 conditions of the years 2030 or 2100, while all other boundary conditions were representative of the year 2000. They found sensitivities of the tropospheric $CH_4$ lifetime of -0.31 ± 0.14 a $K^{-1}$ (-3.2 ± 1.0 % $K^{-1}$) and -0.34 ± 0.12 a $K^{-1}$ (-3.4 ± 0.8 % $K^{-1}$) for the year 2030 and the year 2100 experiments, re-
spectively[1]. The CMIP6 AerChemMIP model ensemble as analyzed by Thornhill et al. (2021a) suggests a sensitivity of the whole-atmosphere $CH_4$ lifetime towards climate change of -0.6 ± 4.5 % $K^{-1}$ assessed from abrupt 4× pre-industrial $CO_2$ experiments. The large intermodel spread results from one model that shows an extension of $CH_4$ lifetime as a result to 4×$CO_2$. The three models showing a shortening of $CH_4$ lifetime suggest a sensitivity of -3.2 ± 0.8 % $K^{-1}$ in close agreement with Voulgarakis et al. (2013).

Our study indicates a larger sensitivity of the $CH_4$ lifetime towards climate change compared to Voulgarakis et al. (2013) and Thornhill et al. (2021a). Possible reasons are the different magnitudes of the perturbations, differences in the simulation set-ups, or the explicit treatment of the $CH_4$ feedback in our study. The similar estimates for the years 2030 and 2100, corresponding to 1.14 K and 4.76 K change of GSAT, respectively[2], by Voulgarakis et al. (2013) suggest that the sensitivity is not highly dependent on the magnitude of the perturbation. Furthermore, the set-ups of individual models in Voulgarakis et al. (2013)
and Thornhill et al. (2021a) differ, e.g. with respect to the level of complexity of the chemical mechanism, whether interactive aerosol is used, or through the different treatment of natural $O_3$ precursor emissions. Nevertheless, the present estimate is larger than the estimates of all individual models in Voulgarakis et al. (2013) and Thornhill et al. (2021a), except for two models which do not parameterize the effect of stratospheric $O_3$ on photolysis rates below, which is taken into account here. In the simulation set-ups analyzed by Voulgarakis et al. (2013) and Thornhill et al. (2021a) $CH_4$ mixing ratios are prescribed at
the lower boundary in all models, except of the GISS-E2-R model analyzed by Voulgarakis et al. (2013). The lifetime response per unit temperature change derived from GISS-E2-R is comparably weak. GISS-E2-R calculates wetland emissions of $CH_4$ online, presumably also for the climate sensitivity experiments, which makes it difficult to compare the $CH_4$ lifetime response in this model to our study. Nevertheless, compared to the other models, the explicit treatment of the $CH_4$ feedback in our set-up allows for a subsequent feedback of OH and correspondingly for a self-feedback on the $CH_4$ lifetime, which can explain the
enhanced sensitivity of the $CH_4$ lifetime towards climate change.

---

[1]Relative estimates were calculated from estimates given in Tables 1 and 4 of Voulgarakis et al. (2013)

[2]Multi-model mean changes of GSAT were calculated from the estimates given in Table 4 of Voulgarakis et al. (2013).



If the $CH_4$ mixing ratio can not adapt to changes in its lifetime, the corresponding $CH_4$ equilibrium mixing ratio can be estimated using Eq. 1, which indicates a global mean $CH_4$ equilibrium mixing ratio in the range of 1.63 to 1.66 parts per million volume (ppmv) for $f$ = [1.2, 1.4] for the present changes of the $CH_4$ lifetime. Thus, Eq. 1 suggests a larger reduction than simulated by the model, which adjusts to a global mean $CH_4$ equilibrium mixing ratio of 1.69 ppmv (see Tab. 2). However,

if the feedback factor is not applied ($f$=1), Eq. 1 gives 1.68 ppmv, which is in close agreement with the simulated response of $CH_4$ mixing ratios and supports the assumption that the sensitivity of OH and the $CH_4$ lifetime towards climate change is larger, if the feedback of $CH_4$ is explicitly simulated as thereby the $CH_4$-OH feedback is implicitly included in the simulated response.

The corresponding response of $O_3$ is shown in Figure 2. The fast response is dominated by $O_3$ increases of up to 8% in the

middle and upper stratosphere. In these regions, $CO_2$ induced stratospheric cooling causes slower chemical $O_3$ depletion (e.g., Rind et al., 1998; Rosenfield et al., 2002; Portmann and Solomon, 2007; Dietmüller et al., 2014; Chiodo et al., 2018). In the lowermost stratosphere, $O_3$ mixing ratios decrease by up to 4%. This decrease can be explained by the so-called reversed self-healing (Rosenfield et al., 2002; Portmann and Solomon, 2007), which describes the effect that increasing $O_3$ above leads to a reduction of ultraviolet radiation that reaches the lower stratosphere and consequently to reduced photochemical production of

$O_3$. The effect of transport from the troposphere into the stratosphere is expected to play a minor role in the fast response, as the strength of tropical up-welling is largely determined by the response of SSTs (Garny et al., 2011; Butchart, 2014). The fast response of tropospheric $O_3$ is smaller than 2%.

The climate response of $O_3$ is dominated by a reduction of up to 10% in the lowermost tropical stratosphere (see Fig. 2 (c)). Enhanced tropical up-welling transports tropospheric $O_3$ poor air from the troposphere into the stratosphere more efficiently.

This is a robust feature across CCMs (Dietmüller et al., 2014; Nowack et al., 2015; Marsh et al., 2016; Chiodo et al., 2018). In the troposphere, $O_3$ mixing ratios decrease by up to 6% in the tropics close to the surface and decrease slightly in the upper tropical troposphere.

The pattern of the full response of stratospheric $O_3$ is qualitatively consistent with previous studies of $O_3$ changes resulting from $CO_2$ perturbation (Dietmüller et al., 2014; Nowack et al., 2015; Marsh et al., 2016; Nowack et al., 2018; Chiodo et al.,

2018; Thornhill et al., 2021a). However, the tropospheric response is different here. Most studies using various CCMs consistently show an increase of $O_3$ in the tropical upper troposphere as part of the full response (Dietmüller et al., 2014; Nowack et al., 2015; Marsh et al., 2016; Nowack et al., 2018; Chiodo et al., 2018). In the studies by Dietmüller et al. (2014) and Nowack et al. (2015, 2018), and presumably also in the studies by Marsh et al. (2016) and Chiodo et al. (2018), $CH_4$ mixing ratios are prescribed at the lower boundary. Consequently, the negative $CH_4$ feedback as discussed above can not evolve. This can lead

to an overestimation of $O_3$ produced from products of the $CH_4$ oxidation and is consistent with the positive response of $O_3$ in the upper tropical troposphere. In particular, the comparison with the study by Dietmüller et al. (2014) indicates an effect of the $CH_4$ feedback on $O_3$ because the EMAC model was used as well in that study.

Using the MESSy submodel TAGGING (Grewe et al., 2017; Rieger et al., 2018) the tropospheric $O_3$ response is attributed to individual source categories representing different processes of $O_3$ production. Consistent with the small fast response of

total tropospheric $O_3$ (Fig. 2 (a)), the corresponding response of the individual categories is below 0.5% of the total reference





O$_3$ for all categories, except for $O_3$ *stratosphere* (see Fig. S4 in the supplement). The response of the category $O_3$ *stratosphere* confirms that less O$_3$ is produced via photolysis in the lower tropical stratosphere. Additionally, it indicates enhanced transport from the stratosphere into the troposphere in middle and higher latitudes of the Northern Hemisphere (NH).

Figure 3 shows the climate response of individual O$_3$ source categories presented as the difference between the fast and full
response of each category in percentage points (p.p.)

$$\Delta O_{3_{\text{cat, climate response}}} = \left( \frac{O_{3_{\text{cat,ECC}}} - O_{3_{\text{cat,REF}}}}{O_{3_{\text{total,REF}}}} \right) - \left( \frac{O_{3_{\text{cat,ERF}}} - O_{3_{\text{cat,REF}}}}{O_{3_{\text{total,REF}}}} \right). \tag{7}$$

The presentation relative to the total reference O$_3$ allows to directly compare the responses of the individual categories to the relative response of total O$_3$ as shown in Fig. 2 (c) and Fig. 3 (a).

The climate response of the category $O_3$ *stratosphere* shows significantly enhanced transport of stratospheric O$_3$ into the
troposphere in both hemispheres. In the extratropical middle troposphere, O$_3$ mixing ratios increase by up to 1.5% relative to the total reference O$_3$ in the full response, which is the largest positive contribution to the total tropospheric O$_3$ response. This supports current knowledge, as enhanced entry of stratospheric O$_3$ under increasing GHG concentration has been a robust feature in CCMs (Abalos et al., 2020). The category $O_3$ *stratosphere* contributes also most to the strong reduction in the lowermost stratosphere. The category $O_3$ *lightning* shows a significant increase of up to 1.25% relative to total reference O$_3$
in the middle tropical troposphere. This is consistent with an increase of lightning NO$_x$ emissions by about 0.3 Tg(N) a$^{-1}$ globally (see Tab. 3). Lightning NO$_x$ is emitted mainly in the upper tropical troposphere where convection is strongest (not shown). In addition, also biogenic emissions of NO$_x$ and C$_5$H$_8$ increase in the full response. Biogenic C$_5$H$_8$ emissions increase strongest in the Amazon region and the Congo river basin, whereas biogenic NO$_x$ emissions increase over land in the tropics and mid latitudes (not shown). However, the zonal mean climate response of $O_3$ *biogenic* is mostly not significant due to
competing effects of enhanced precursor emissions and of enhanced chemical loss with H$_2$O. An enhanced sink of O$_3$ via the reaction of O($^1$D) with H$_2$O, which leads to effective O$_3$ loss, is expected in a warmer and moister troposphere (e.g., Stevenson et al., 2006). The spatial distribution of the tropospheric O$_3$ column shows mainly a reduction over the tropical ocean (see Fig. S6 in the supplement), which is also reflected by the significant reduction between the equator and 30° N in the zonal mean (Fig. 3 (d)). Locally over regions with increasing precursor emissions, e.g. over the Amazon region and the Congo river
basin, the tropospheric O$_3$ column increases in the category $O_3$ *biogenic* (see Fig. S6 in the supplement). Anthropogenic and biomass burning emissions are prescribed and are therefore not affected by the CO$_2$ increase. In these categories, decreasing O$_3$ from enhanced loss or reduced O$_3$ production efficiency is shown. The reduction of $O_3$ *anthropogenic* is most pronounced over the tropical ocean, where a decline of O$_3$ due to enhanced loss via H$_2$O is expected (Stevenson et al., 2006; Zanis et al., 2022). The effect of the reduction of $O_3$ *biomass burning* on total O$_3$ is small, since also its absolute contribution is small.
In addition to the enhanced sink, reduced O$_3$ production per emitted NO$_x$ could play a role in the latter two categories as O$_3$ precursor emissions from natural categories increase. The category $O_3$ *CH$_4$* shows a reduction throughout the troposphere. This is consistent with the reduction of CH$_4$ mixing ratios, as in the new equilibrium fewer products of the CH$_4$ oxidation are available for O$_3$ production resulting in reduced O$_3$ production in this category. Further, enhanced chemical loss can contribute to the reduction of this category. In the upper tropical troposphere the increase of lightning NO$_x$ emissions counteracts the





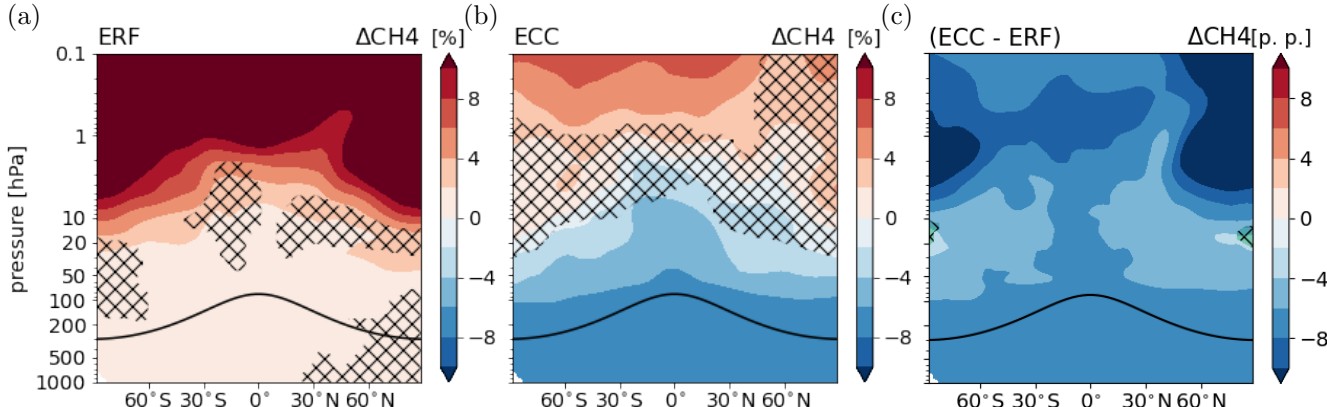

**Figure 1.** CH$_4$ response following the CO$_2$ perturbation: Relative differences between the annual zonal mean CH$_4$ mixing ratios of sensitivity simulations (a) ERFCO$_2$ (fast response) and (b) ECCCO$_2$ (full response) and their respective reference simulation in [%]. (c) Climate response as difference between the CH$_4$ responses in panels (a) and (b) in [percentage points (p.p.)]. Non-hatched areas are significant on the 95% confidence level according to a Welch's test based on annual mean values. The solid black line indicates the location of the climatological tropopause.

**Table 2.** Global annual mean values of tropospheric CH$_4$ lifetime with respect to the oxidation with OH, and CH$_4$ surface volume mixing ratio for the performed simulations.

|  | Trop. CH$_4$ lifetime [a] | CH$_4$ surface VMR [ppmv] |
|---|---|---|
| REF-SSTfix | $7.59 \pm 0.03$ | $1.82 \pm 0.00$ |
| REF-SSTvar | $7.58 \pm 0.03$ | $1.82 \pm 0.00$ |
| ERFCO$_2$ | $7.59 \pm 0.03$ | $1.82 \pm 0.00$ |
| ECCCO$_2$ | $7.02 \pm 0.05$ | $1.69 \pm 0.00$ |
| ERFCH$_4$ | $14.48 \pm 0.04$ | $8.66 \pm 0.01$ |
| ECCCH$_4$ | $13.20 \pm 0.08$ | $8.05 \pm 0.01$ |

The values after the $\pm$ are the corresponding interannual standard deviations based on 20
annual mean values, which are listed to estimate the year to year variability.

effect of the CH$_4$ decrease by providing enhanced levels of NO$_x$, which react with the products of the CH$_4$ oxidation more efficiently. The climate response of $O_3$ $CH_4$ is not significant in this region. The climate response in the category $O_3$ $N_2O$ shows significant decreases in the lower stratosphere and troposphere. In the stratosphere, N$_2$O mixing ratios increase (not shown) indicating less N$_2$O decomposition (Dietmüller et al., 2014). Thereby, less nitrogen oxide (NO) is produced to form O$_3$, which is consistent with the decrease of O$_3$ formed from N$_2$O decomposition.



**Table 3.** Global annual mean emissions of $NO_x$ from lightning, $NO_x$ from biogenic sources, and $C_5H_8$ from biogenic sources.

|  | Lightning $NO_x$ [Tg(N) a$^{-1}$] | Biogenic $NO_x$ [Tg(N) a$^{-1}$] | Biogenic $C_5H_8$ [Tg(C) a$^{-1}$] |
|---|---|---|---|
| REF-SSTfix | $5.2 \pm 0.1$ | $6.0 \pm 0.0$ | $307 \pm 3$ |
| REF-SSTvar | $5.2 \pm 0.1$ | $6.0 \pm 0.0$ | $306 \pm 4$ |
| ERFCO$_2$ | $5.3 \pm 0.1$ | $6.0 \pm 0.0$ | $307 \pm 4$ |
| ECCCO$_2$ | $5.6 \pm 0.1$ | $6.4 \pm 0.0$ | $338 \pm 6$ |
| ERFCH$_4$ | $4.9 \pm 0.1$ | $6.0 \pm 0.0$ | $306 \pm 3$ |
| ECCCH$_4$ | $5.1 \pm 0.1$ | $6.4 \pm 0.0$ | $337 \pm 5$ |

The values after the $\pm$ are the corresponding interannual standard deviations based on 20 annual mean values, which are listed to estimate the year to year variability. The model calculated biogenic $C_5H_8$ emissions are scaled by a factor of 0.6 before added to the atmospheric $C_5H_8$ tracer (see Jöckel et al., 2016). The values shown here include the scaling.

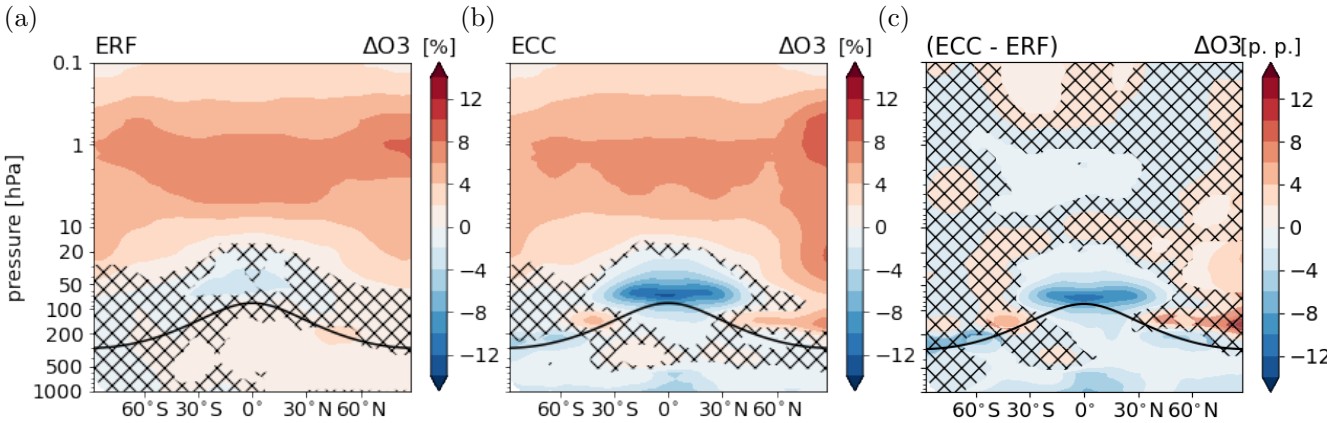

**Figure 2.** Same as Fig. 1, but for $O_3$.

## 3.2 Methane and ozone composition changes following $2.75 \times CH_4$ emission flux perturbation

In this section, we present the simulation results of the $2.75 \times CH_4$ emission flux perturbation. Figure 4 shows the zonal mean distribution of $CH_4$ mixing ratios of the reference simulation REF-SSTfix and of the two simulations with $CH_4$ emission fluxes increased by a globally constant factor of 2.75. As expected, $CH_4$ mixing ratios increase everywhere in the fast response shown in Fig. 4 (b). Hereby, the increase factor of $CH_4$ mixing ratios is even larger than the increase factor of the emission fluxes. Tab. 2 shows that an increase of $CH_4$ emissions by a factor of 2.75 results in an increase of the global mean surface $CH_4$ mixing ratio by a factor of 4.76. This is caused by a large extension of the tropospheric $CH_4$ lifetime by about 7 years (see Tab. 2). The $CH_4$ increase reduces the tropospheric OH mixing ratios by up to 60% (see Fig. S3 in the supplement), thereby extending the $CH_4$ lifetime.





**Figure 3.** Climate response of tropospheric $O_3$ following the $CO_2$ perturbation: (a) response of total $O_3$ (same as Fig. 2 (c), but differently scaled color levels to better compare with the response in the individual source categories), (b) - (h) response of $O_3$ in individual categories relative to total reference $O_3$ ($\Delta O_{3\text{cat, climate response}} = (\frac{O_{3\text{cat,ECC}} - O_{3\text{cat,REF}}}{O_{3\text{total,REF}}}) - (\frac{O_{3\text{cat,ERF}} - O_{3\text{cat,REF}}}{O_{3\text{total,REF}}})$). Non-hatched areas are significant on the 95% confidence level according to a Welch's test based on annual mean values. The solid black line indicates the location of the climatological tropopause.

A similar effect was found by Winterstein et al. (2019) who analyzed the fast response of 2× and 5×$CH_4$ surface mixing ratios in a set-up with prescribed $CH_4$ surface mixing ratios also using EMAC. The magnitude of the present $CH_4$ perturbation is comparable to their 5×$CH_4$ experiment. In particular, to reach the prescribed $CH_4$ surface mixing ratios a pseudo surface



emission flux is diagnosed in their set-up. The increase factor of the pseudo flux that corresponds to an increase of $5\times CH_4$ is 2.75 (Stecher et al., 2021), exactly the increase factor of $CH_4$ surface emissions used in our study. Thus, in our study the increase of emission fluxes results in a close to fivefold increase of the $CH_4$ surface mixing ratio. The global mean reference

$CH_4$ mixing ratio and the corresponding pseudo emission flux are slightly lower in Winterstein et al. (2019), namely about 1.8 ppmv and 567.7 Tg($CH_4$) a$^{-1}$. Additionally, the spatial distribution of their diagnosed pseudo flux is different and might be unrealistic. The latter two points explain, why the increase factor of $CH_4$ mixing ratios is not exactly the same in our study. Nevertheless, the results suggest that the relation between the increase of $CH_4$ emissions and mixing ratios at the lower boundary is consistent, if either the emissions or the mixing ratios are increased.

In the full response, as shown in Fig. 4 (c), $CH_4$ mixing ratios are lower in comparison to the fast response, as shown in Fig. 4 (b). Similar as in the climate response following the $CO_2$ perturbation, higher tropospheric temperatures lead to increased production of OH (see Fig. S3 in the supplement). Additionally, the temperature dependent reaction rate coefficient leads to a faster $CH_4$ oxidation. The corresponding sensitivity of the $CH_4$ lifetime per unit change of GSAT is -1.09 a K$^{-1}$ or -7.6% K$^{-1}$ (relative to the lifetime in ERFCH$_4$). Both, the absolute and the relative sensitivity, are larger compared to the $CO_2$ perturbation

experiment, which is possibly caused by the different $CH_4$ conditions in the respective fast responses (ERFCO$_2$ and ERFCH$_4$).

Stecher et al. (2021) analyzed the climate response of the $2\times$ and $5\times$ $CH_4$ surface mixing ratio experiments corresponding to Winterstein et al. (2019). The sensitivity of $CH_4$ lifetime per unit change of GSAT corresponding to their $5\times CH_4$ surface mixing ratio perturbation is $\frac{-1.17\,\text{a}}{1.28\,\text{K}} = -0.91$ a K$^{-1}$, which corresponds to a relative change of -5.9% K$^{-1}$ (relative to the lifetime in their fast response), and is thus less pronounced than the respective value of our study. The major difference

between the simulation set-ups is that $CH_4$ mixing ratios can not respond to the lifetime response in the set-up of Stecher et al. (2021). This suggests that in the present study the sensitivity of $CH_4$ lifetime towards climate change is enhanced, because the $CH_4$-OH feedback is included in the response of OH and, therefore, in the $CH_4$ lifetime. The same is indicated by the results of the $CO_2$ perturbation as discussed above.

The response of $O_3$ is shown in Fig. 5. In the fast response (panel (a)), $O_3$ mixing ratios increase significantly throughout the

troposphere with a maximum increase of up to 60% in the upper tropical troposphere. The $CH_4$ perturbation leads to enhanced $O_3$ formation by enhanced production of $O_3$ precursor species through $CH_4$ oxidation. In the stratosphere, radiatively induced cooling (a rapid adjustment) leads to $O_3$ increases in the middle stratosphere. Above 1 hPa, $O_3$ mixing ratios decrease due to enhanced catalytic depletion by odd hydrogen (HO$_x$). HO$_x$ is increased by enhanced production of stratospheric $H_2O$ caused by the $CH_4$ oxidation (see Fig. S2 in the supplement) and also by enhanced formation via the sink reaction of $CH_4$ with O($^1$D).

In the lower tropical stratosphere, $O_3$ decreases, which can be explained by the reversed self-healing effect (Rosenfield et al., 2002; Portmann and Solomon, 2007), which is also effective for the $CO_2$ perturbation (see above). The fast response of $O_3$ is consistent with the fast response evolving in the comparable $5\times CH_4$ surface mixing ratio experiment (Winterstein et al., 2019), as the same processes are effective, which are explained in more detail by Winterstein et al. (2019).



Fig. 6 shows the fast response of individual $O_3$ categories derived using the TAGGING submodel. Shown is the difference
between ERFCH$_4$ and REF-SSTfix of one category relative to the total reference $O_3$

$$\Delta O_{3_{\text{cat}}} = \frac{O_{3_{\text{cat,ERF}}} - O_{3_{\text{cat,REF}}}}{O_{3_{\text{total,REF}}}},$$

allowing a direct comparison with the relative response of total $O_3$. The $O_3$ mixing ratios increase in all categories except for
the category *$O_3$ stratosphere*. This category shows reduced $O_3$ production through photolysis of $O_2$ in the lower stratosphere
consistent with the reverse self-healing effect. The increase is strongest in the category *$O_3$ CH$_4$*, as the CH$_4$ increase directly
leads to the formation of $O_3$. The increase of this category is most pronounced in the upper tropical troposphere and reaches
up to 30% relative to the total reference $O_3$. The larger abundance of NMHCs and CO also affects $O_3$ production of the other
categories as their reaction with precursors from other categories, in particular NO$_x$, leads to enhanced $O_3$ production in the
category *$O_3$ CH$_4$*, but also in the other categories. This effect is largest for the category *$O_3$ lightning*, which shows $O_3$ increases
of up to 20% relative to the total reference $O_3$, even though emissions of lightning NO$_x$ decrease by 0.3 Tg(N) a$^{-1}$ globally
in the simulation ERFCH$_4$ compared to REF-SSTfix (see Tab. 3). The CH$_4$ perturbation leads to upper tropospheric/lower
stratospheric warming peaking at around 100 hPa in the tropics (see Fig. S1 in the supplement). The higher static stability
leads to less convection and thereby to decreasing lightning NO$_x$ emissions. Upper troposphere/lower stratosphere warming
following increased CH$_4$ has been already noted elsewhere and is expected to be even more pronounced if shortwave (SW)
absorption by CH$_4$ is accounted for in the simulation set-up (Modak et al., 2018; Allen et al., 2023). Nevertheless, the enhanced
abundance of precursors from CH$_4$ oxidation lead to enhanced $O_3$ production in this category. The category showing the third
most pronounced increase is *$O_3$ anthropogenic*. Here, the increase relative to the total reference $O_3$ is with up to 15% most
pronounced in the lower NH, where the contribution of *$O_3$ anthropogenic* to total $O_3$ is largest.

The climate response of $O_3$ shown as the difference between full and fast response (see Fig. 5 (c)) represents the isolated
effect of the GSAT response on $O_3$. It shows a strong reduction of $O_3$ mixing ratios in the lower tropical stratosphere, which
is caused by enhanced tropical up-welling (consistent with the CO$_2$ simulation). In the Northern polar lower stratosphere,
$O_3$ mixing ratios are enhanced pointing towards strengthened poleward and downward transport (i.e. strengthening of the
Brewer-Dobson circulation) of stratospheric air masses. In the Southern polar tropopause region the rise of the tropopause in
the climate response leads to a large $O_3$ reduction. This process is also apparent in the NH, albeit less pronounced. Apart from
that, the full response of $O_3$ in the stratosphere is mainly caused by the fast response. The climate response of tropospheric
$O_3$ shows a reduction, except for the tropical middle troposphere, where the response shows a weak, not significant, increase
in the zonal mean. In this region, enhanced emissions of lightning NO$_x$ lead to enhanced $O_3$ formation (see also discussion of
TAGGING results below). The climate response of $O_3$ is strikingly similar with the climate response pattern resulting from the
CO$_2$ perturbation (see Fig. 2 (c)), even though the fast response is different.

The similarity of the climate response patterns of $O_3$ resulting from CO$_2$ and CH$_4$ perturbations has been also noted by
Stecher et al. (2021). However, the $O_3$ climate response resulting from their 5×CH$_4$ mixing ratio increase shows a significant
increase of $O_3$ in the tropical middle troposphere. As already stated, the main difference between their set-up and ours is that




the feedback of $CH_4$ mixing ratios, and thereby also any secondary effects of $O_3$, is suppressed. As for the $CO_2$ perturbation, the reduction of $CH_4$ mixing ratios feeds back on $O_3$.

Fig. 7 shows the climate response of individual $O_3$ source categories. It is calculated using Eq. 7 in accordance with the
corresponding analysis of the $CO_2$ perturbation. The patterns of the climate response of the individual categories are overall consistent with those of the $CO_2$ perturbation.

The category *$O_3$ stratosphere* increases in the troposphere, indicating enhanced transport of stratospheric $O_3$ into the troposphere. The increase is significant everywhere, except for the extratropical Southern Hemisphere (SH) and the lower tropical troposphere. This category contributes strongest to the reduction of $O_3$ in the lower tropical stratosphere. The category *$O_3$*
*lightning* shows increases in the tropical middle troposphere resulting from an increase of the lightning $NO_x$ emissions by 0.2 Tg(N) a$^{-1}$ in ECCCH$_4$ compared to ERFCH$_4$ (see Tab. 3) and decreases in the lower troposphere. Biogenic $NO_x$ emissions increase by 0.37 Tg(N) a$^{-1}$ and biogenic $C_5H_8$ emissions increase by about 30 Tg(C) a$^{-1}$ as reaction to climate change (see Tab. 3). However, the zonal mean climate response of $O_3$ in this category is mostly not significant and shows a decrease in the lower tropical and upper NH troposphere (see Fig. 7 (d)). The tropospheric $O_3$ columns in this category increase lo-
cally over the Amazon region and the Congo river basin, where biogenic emissions of $C_5H_8$ increase strongest, and decrease mostly over the tropical ocean (see Fig. S8 in the supplement). As already mentioned, the sink of $O_3$ via the reaction of $O(^1D)$ with $H_2O$ is expected to strengthen in a warmer and moister troposphere. Similar as for the $CO_2$ perturbation, the effects of increased $O_3$ precursor emissions and the enhanced chemical sink due to a larger abundance of tropospheric $H_2O$ compete in this category. The category *$O_3$ CH$_4$* decreases everywhere in the zonal mean, except for the tropical middle troposphere. The
reduction is consistent with the reduction of $CH_4$ mixing ratios in the climate response, which leads to a reduced formation of $O_3$. In addition, the enhanced chemical sink leads to further reduction of $O_3$. The increase in the tropical middle troposphere coincides with the maximum increase of $O_3$ production from lightning $NO_x$ emissions, which indicates that enhanced $NO_x$ from lightning reacts with products of the $CH_4$ oxidation resulting in an increased $O_3$ production in both categories. The corresponding response of the tropospheric $O_3$ columns is not significant in the tropics, because of the counteracting responses
in the lower and middle troposphere, but shows a significant decrease in the extra-tropics (see Fig. S8 in the supplement). The categories with prescribed $O_3$ precursor emissions, *$O_3$ biomass burning* and *$O_3$ anthropogenic*, show decreased $O_3$ mixing ratios throughout the troposphere (see Fig. 7 (g) and (h)), consistently with the climate response resulting from the $CO_2$ perturbation. Additionally, reduced $O_3$ production per emitted molecule $NO_x$ can play a role as $O_3$ precursor emissions of natural categories increase.




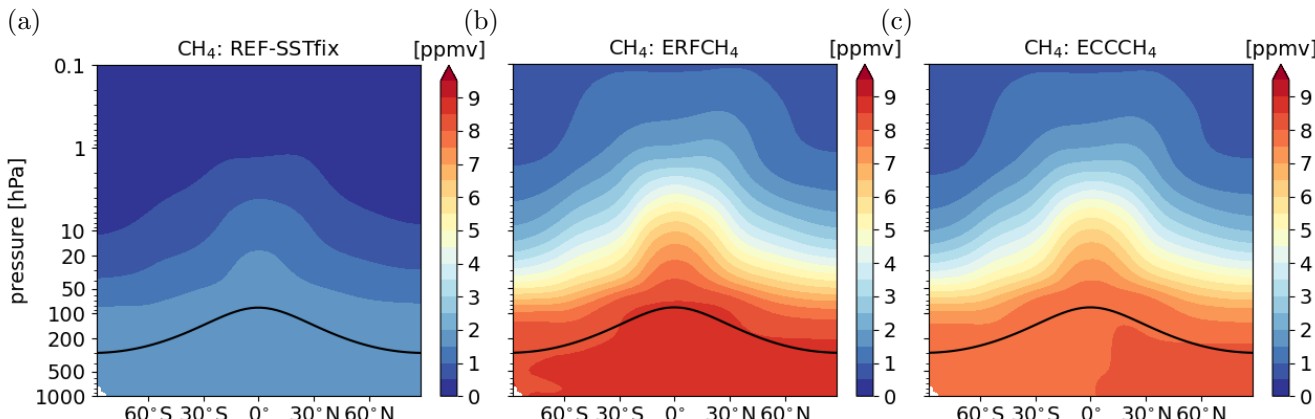

**Figure 4.** Annual zonal mean distribution of CH$_4$ mixing ratios in simulation (a) REF-SSTfix, (b) ERFCH$_4$ (fast response) and (c) ECCCH$_4$ (full response) in [ppmv].

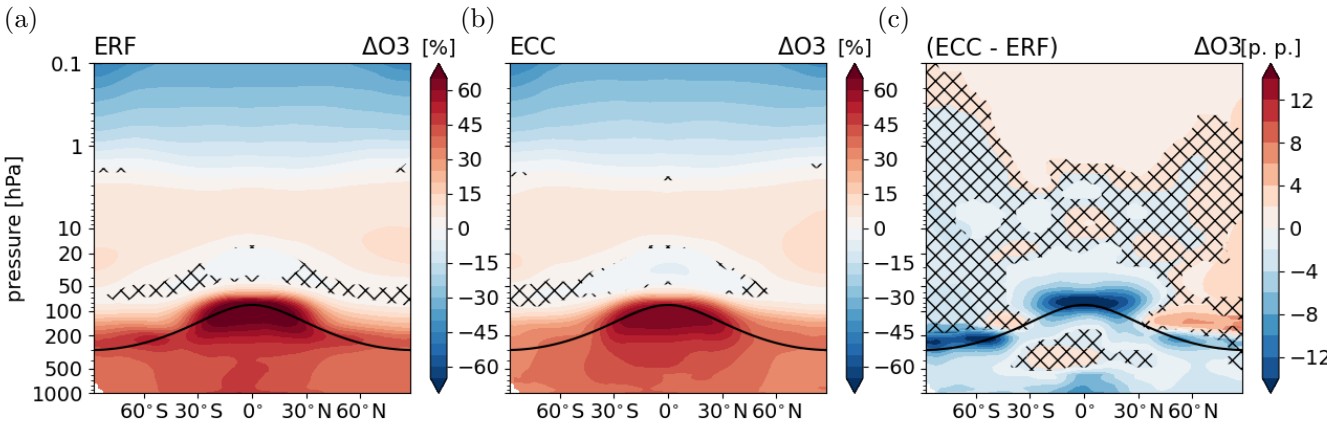

**Figure 5.** O$_3$ response following the CH$_4$ perturbation (same as Fig. 2, but for the CH$_4$ perturbation): Relative differences between the annual zonal mean O$_3$ mixing ratios of sensitivity simulations (a) ERFCH$_4$ (fast response) and (b) ECCCH$_4$ (full response) and their respective reference simulation in [%]. (c) Climate response as difference between the O$_3$ responses in panels (a) and (b) in [percentage points (p.p.)]. Non-hatched areas are significant on the 95% confidence level according to a Welch's test based on annual mean values. The solid black line indicates the location of the climatological tropopause.




**Figure 6.** Fast response of tropospheric $O_3$ following the $CH_4$ perturbation: (a) response of total $O_3$ (same as Fig. 4 (a), but differently scaled colour levels to better compare with the response in the individual categories), (b) - (h) response of $O_3$ in individual source categories relative to total reference $O_3$ ($\Delta O_{3_{\text{cat}}} = \frac{O_{3_{\text{cat,ERF}}} - O_{3_{\text{cat,REF}}}}{O_{3_{\text{total,REF}}}}$). Non-hatched areas are significant on the 95% confidence level according to a Welch's test based on annual mean values. The solid black line indicates the location of the climatological tropopause.



**Figure 7.** Climate response of tropospheric $O_3$ following the $CH_4$ perturbation (same as Fig. 3, but for the $CH_4$ perturbation): (a) response of total $O_3$ (same as Fig. 5 (c), but differently scaled colour levels to better compare with the response in the individual categories), (b) - (h) response of $O_3$ in individual source categories relative to total reference $O_3$ ($\Delta O_{3_{\text{cat, climate response}}} = \left(\frac{O_{3_{\text{cat,ECC}}} - O_{3_{\text{cat,REF}}}}{O_{3_{\text{total,REF}}}}\right) - \left(\frac{O_{3_{\text{cat,ERF}}} - O_{3_{\text{cat,REF}}}}{O_{3_{\text{total,REF}}}}\right)$). Non-hatched areas are significant on the 95% confidence level according to a Welch's test based on annual mean values. The solid black line indicates the location of the climatological tropopause.



## 3.3 Radiative effects and climate sensitivity

Table 4 shows results for the total SARF, ERF, $\Delta$GSAT and the associated climate sensitivity parameters $\lambda$, as well as individual radiative effects corresponding to the composition changes of $CH_4$, $O_3$ and stratospheric $H_2O$.

For the $CO_2$ perturbation, the estimate of ERF is smaller than SARF, but the difference is not significant due to the large statistical uncertainty associated with ERF (e.g, Forster et al., 2016). Along with this, the climate sensitivity parameter based on ERF, $\lambda_{ERF}$, is larger compared to $\lambda_{SARF}$, but the difference is not statistically significant either. In contrast, for the $CH_4$ perturbation, the estimate of ERF of $1.72\pm0.17$ W m$^{-2}$ is considerably larger than SARF, which is estimated at 0.51 W m$^{-2}$. The estimate of SARF reproduces the result of the comparable $5\times CH_4$ mixing ratios experiment with the EMAC model (Winterstein et al., 2019). However, the radiative effect of $CH_4$ is known to be underestimated by the used radiative transfer scheme (Winterstein et al., 2019; Nützel et al., 2024). For instance, using the formula by Etminan et al. (2016)[3] for the present $CH_4$ perturbation a SARF of about 1.7 W m$^{-2}$ is derived. The underestimation of the radiative effect of $CH_4$ affects the estimate of ERF as well. Under the assumption that all other adjustments remain the same, only the direct contribution of $CH_4$ is exchanged, which results in an ERF of about 2.9 W m$^{-2}$. This considerably larger ERF suggests a correspondingly larger response of GSAT as well.

As illustrated by Tab. 4, chemical adjustments play a minor role for the $CO_2$ perturbation. The fast response of stratospheric $O_3$ induces a negative adjustment of -0.034 W m$^{-2}$, whereas the fast response of tropospheric $O_3$ induces a positive adjustment of 0.012 W m$^{-2}$. The adjustment of tropospheric $CH_4$ is negligible, whereas stratospheric $CH_4$ induces a small positive adjustment due to its increase in the stratosphere and associated local radiative cooling. $H_2O$ increases in the lowermost stratosphere by up to 5% (see Fig. S2 in the supplement), which leads to a positive adjustment of 0.015 W m$^{-2}$. The change of $H_2O$ in the lowermost stratosphere is largely driven by a change of the tropical cold point temperature (CPT) (see Fig. S9). For the $CO_2$ perturbation, chemical production of $H_2O$ in the stratosphere is slightly reduced up to about 2 hPa. Thus, the adjustment of stratospheric $H_2O$ is unlikely to be chemically induced. To summarize, interactive chemistry dampens the ERF of the $CO_2$ perturbation by about -1.3%, mainly by the effect of stratospheric $O_3$.

For the $CH_4$ perturbation, chemical adjustments of $O_3$ and stratospheric $H_2O$ are important contributions to the ERF. The adjustment of tropospheric $O_3$ is 0.64 W m$^{-2}$, and of stratospheric $O_3$ it is 0.16 W m$^{-2}$. In addition, the adjustment of stratospheric $H_2O$ is estimated at 0.51 W m$^{-2}$. The $CH_4$ perturbation leads to relative increases of $H_2O$ up to 250% in the upper stratosphere and mesosphere (see Fig. S2 in the supplement) because the increased abundance of $CH_4$ leads to enhanced production of $H_2O$ by the $CH_4$ oxidation. Additionally, warming of the tropical cold point (see Fig. S9) leads to reduced dehydration of upwelling air parcels, and thus to an increased abundance of $H_2O$ in the lower stratosphere. The zonal mean warming of the tropical cold point is 1.5 K and thereby more pronounced than in the respective $CO_2$ experiment. The $CH_4$ perturbation induces a direct radiative heating at the tropical cold point of up to 1 K (see Fig. S11 in the supplement), and the response of stratospheric $O_3$ leads to additional radiative heating of about 1.5 K in this region (see Fig. S11 in the supplement).

---

[3]The $CH_4$ mixing ratios of simulation ERFCH4 are outside of the range tested to derive the formula of Etminan et al. (2016), but the formula can still provide a rough estimate for the $CH_4$ radiative effect.



To summarize, the adjustments of $O_3$ and stratospheric $H_2O$ enhance the ERF of the $CH_4$ perturbation by 1.31 W m$^{-2}$. The estimates of the adjustments of $O_3$ and stratospheric $H_2O$ are consistent with the results of the comparable 5×$CH_4$ mixing ratios experiment (Winterstein et al., 2019, see also column 3 in Tab. 4).

The feedback parameters represent the radiative effects induced by composition changes of the climate response, i.e. caused by the isolated effect of GSAT changes, normalized by the corresponding GSAT response. For both perturbation types, $CO_2$ and $CH_4$, the feedbacks of $CH_4$ and $O_3$ are negative, which means that they dampen the resulting temperature change. The feedback parameter corresponding to the $CH_4$ reduction in the $CO_2$ perturbation experiment is -0.025 W m$^{-2}$ K$^{-1}$. As already mentioned, it is known that the direct radiative effect of $CH_4$ is underestimated by the used radiative transfer scheme (Winter-

stein et al., 2019; Nützel et al., 2024). Therefore, we additionally calculate the feedback parameter for the same change of $CH_4$ mixing ratios, but with the PSrad radiation scheme (Pincus and Stevens, 2013; Nützel et al., 2024), using otherwise the same methodology (see Sect. 2.3). With PSrad, the $CH_4$ feedback is estimated at -0.041 W m$^{-2}$ K$^{-1}$ implying a more pronounced negative radiative feedback. Applying the formula of Etminan et al. (2016) for the change of $CH_4$ mixing ratio diagnosed from the simulation suggests a radiative effect of -0.059 W m$^{-2}$, which corresponds to a feedback parameter of -0.054 W m$^{-2}$ K$^{-1}$.

Previous estimates of the $CH_4$ feedback have been derived offline from the change of atmospheric $CH_4$ lifetime, and range from -0.014±0.067 W m$^{-2}$ K$^{-1}$ (Thornhill et al., 2021a, if the estimates for changes of biogenic volatile organic compounds, lightning $NO_x$ and meteorology are combined[4]), -0.03±0.01 W m$^{-2}$ K$^{-1}$ (Heinze et al., 2019) to -0.036 W m$^{-2}$ K$^{-1}$ (Diet-müller et al., 2014). Our estimate using the PSrad scheme is at the upper end of previous estimates, but in the range of estimates from individual models analyzed by Thornhill et al. (2021a).

For the $CH_4$ perturbation, the feedback associated with the reduction of $CH_4$ mixing ratios in the full response in comparison to the fast response is -0.019 W m$^{-2}$ K$^{-1}$. Using the PSrad radiation scheme with otherwise the same method, the feedback parameter is estimated at -0.089 W m$^{-2}$ K$^{-1}$ suggesting a clearly larger influence. The radiative feedback of $CH_4$ corresponding to the 5× $CH_4$ mixing ratio experiment (Winterstein et al., 2019; Stecher et al., 2021) does not include the reduction of $CH_4$ in the troposphere. The corresponding feedback parameter indicates a small positive feedback, caused by the larger mixing ratios

of $CH_4$ in the stratosphere in the full response compared to the fast response (Stecher et al., 2021).

The feedback parameters corresponding to $O_3$ changes in the troposphere and stratosphere are both negative and add to total feedback parameters of -0.039 W m$^{-2}$ K$^{-1}$ for the $CO_2$ perturbation, and -0.054 W m$^{-2}$ K$^{-1}$ for the $CH_4$ perturbation. Previous studies of the $O_3$ feedback resulting from $CO_2$ perturbations have assessed the full response in contrast to the climate response. The feedback parameter corresponding to the full response, i.e. including the adjustment, is -0.059 W m$^{-2}$ K$^{-1}$ in our

study. Previous estimates range from -0.015 W m$^{-2}$ K$^{-1}$ and -0.022 W m$^{-2}$ K$^{-1}$ (Dietmüller et al., 2014), -0.018 W m$^{-2}$ K$^{-1}$ (Marsh et al., 2016), -0.046±0.018 W m$^{-2}$ K$^{-1}$ (Thornhill et al., 2021a), to -0.12 W m$^{-2}$ K$^{-1}$ (Nowack et al., 2015, if a corresponding GSAT response of 5.75 K is assumed). The feedback parameter of total $O_3$ in the present study lies in the range of previous estimates, but, notably, is more pronounced than the estimate by Dietmüller et al. (2014), who also used the EMAC model. Part of the difference can be explained by the different sign of the feedback of tropospheric $O_3$. Dietmüller et al.

---

[4]If the model CESM2-WACCM, which projects a prolongation of $CH_4$ lifetime with climate change, is excluded, the $CH_4$ feedback is estimated at -0.053±0.010 W m$^{-2}$ K$^{-1}$. The given uncertainties are standard deviations across models.



(2014) found a positive feedback parameter of 0.008 W m$^{-2}$ K$^{-1}$ to 0.009 W m$^{-2}$ K$^{-1}$ for tropospheric $O_3$ compared to the negative feedback parameter in this study. The reduction of tropospheric $CH_4$ mixing ratios leads to reduced $O_3$ production and thereby modifies the response of $O_3$ as discussed above. This indirect effect on $O_3$ is a consequence of applying emission fluxes instead of a prescribed lower boundary mixing ratio for $CH_4$. In addition, also the negative feedback of stratospheric $O_3$ is more pronounced in this study, which might be explained by the different magnitude of the perturbations. Dietmüller et al.

(2014) noted differences between their 2× and 4×$CO_2$ experiments. Therefore, deviations for the 1.35×$CO_2$ in this study can be expected. In addition, the different vertical resolution of the simulation set-ups might affect the response of stratospheric $O_3$. The results of the 5×$CH_4$ mixing ratio experiment by Stecher et al. (2021) do not indicate a significant feedback of total $O_3$. This suggests that the reduction of $CH_4$ mixing ratios in the full response drives the negative $O_3$ feedback.

     As explained above, the scaling factors of the $CO_2$ and $CH_4$ increase are chosen so that the resulting ERFs are comparable

to allow an optimal comparison of the climate sensitivity of the two perturbation types, as the latter can depend on the magnitude of the radiative perturbation (e.g., Hansen et al., 2005; Dietmüller et al., 2014). The estimates of the climate sensitivity parameter based on ERF, $\lambda_{ERF}$, are identical for the $CO_2$ and $CH_4$ perturbations of this study. In contrast, the climate sensitivity parameters based on SARF, $\lambda_{SARF}$, differ significantly between the $CH_4$ and the $CO_2$ perturbation. This finding confirms the results of previous studies that the climate sensitivity is in general less dependent on the type of perturbation for ERF (e.g.,

Richardson et al., 2019). The use of $\lambda_{ERF}$ as climate sensitivity parameter to obtain an efficacy close to unity for the $CH_4$ perturbation is more important in this study compared to Richardson et al. (2019), because of the effect of interactive chemistry, which leads to larger differences between SARF and ERF for the $CH_4$ perturbation.

     The estimate of $\lambda_{ERF}$ of 0.68±0.08 K / (W m$^{-2}$) is smaller than the corresponding estimate of the 5×$CH_4$ mixing ratio experiment of 0.72±0.07 K / (W m$^{-2}$) (Stecher et al., 2021). This suggests a reduction of the climate sensitivity parameter

caused by the explicit simulation of the $CH_4$ reduction which would be physically consistent with the negative feedbacks of $O_3$ and $CH_4$ in this study. The difference between the two estimates is, however, not statistically significant.

## 4   Discussion and Conclusions

In this study, we assess the feedback of atmospheric $CH_4$ resulting from changes of its chemical sink, which is mainly the oxidation with OH and, which is influenced by temperature and the chemical composition of the atmosphere. We present

results from numerical simulations with the CCM EMAC perturbed by either 1.35×$CO_2$ mixing ratio or 2.75×$CH_4$ emission flux increase. The scaling factors were chosen to reach a comparable ERF for both perturbation agents. EMAC is used in a $CH_4$ emission flux driven setup, which allows the atmospheric $CH_4$ mixing ratio to adjust to changes of the chemical sink without constraints.

     The increase of $CH_4$ emissions by a globally constant factor of 2.75 corresponds to an increase of the global mean $CH_4$

surface mixing ratio by a factor of 4.76. The larger increase of the $CH_4$ mixing ratio compared to the emissions is caused by a strong reduction of tropospheric OH, which leads to the extension of the tropospheric $CH_4$ lifetime. A similar effect was found by Winterstein et al. (2019), who analyzed the response of 5×$CH_4$ mixing ratios using a comparable set-up of the EMAC





**Table 4.** Estimates of total SARF, ERF, $\Delta$GSAT and the corresponding climate sensitivity parameters $\lambda$, as well as adjustments and feedbacks of individual composition changes of $CH_4$, $O_3$ and stratospheric $H_2O$. The climate sensitivity parameters $\lambda_{SARF}$ and $\lambda_{ERF}$ are calculated using SARF or ERF, respectively. Adjustments are calculated as the radiative flux changes of the fast response (in W m$^{-2}$). Feedbacks are calculated as the difference of radiative flux changes between the full and the fast response divided by the corresponding change of global surface air temperature $\Delta$GSAT (in W m$^{-2}$ K$^{-1}$). The radiative effects of individual composition changes include the corresponding stratospheric temperature adjustment (Stuber et al., 2001). All radiative estimates are evaluated at TOA. In addition, the estimates of the 5$\times CH_4$ volume mixing ratio experiments analyzed by Winterstein et al. (2019) and Stecher et al. (2021) are shown in the third column.

| Perturbation | | **1.35$\times CO_2$ VMR** | **2.75$\times CH_4$ emissions** | **5$\times CH_4$ VMR** |
|---|---|---|---|---|
| | | ERFCO2 / ECCCO2 | ERFCH4 / ECCCH4 | (Winterstein et al., 2019; Stecher et al., 2021) |
| SARF | [W m$^{-2}$] | 1.71 | 0.51 | 0.51 |
| ERF | [W m$^{-2}$] | 1.61$\pm$0.16 | 1.72$\pm$0.17 | 1.79$\pm$0.17 |
| $\Delta$GSAT | [K] | 1.09$\pm$0.06 | 1.17$\pm$0.06 | 1.28$\pm$0.04 |
| $\lambda_{SARF}$ | [K / W m$^{-2}$] | 0.64$\pm$0.03 | 2.30$\pm$0.11 | 2.49$\pm$0.08 |
| $\lambda_{ERF}$ | [K / W m$^{-2}$] | 0.68$\pm$0.07 | 0.68$\pm$0.08 | 0.72$\pm$0.07 |
| Adjustments | [W m$^{-2}$] | | | |
| $O_3$ trop. | | 0.012 | 0.64 | 0.56 |
| $O_3$ strat. | | -0.034 | 0.16 | 0.20 |
| $O_3$ total | | -0.022 | 0.81 | 0.76 |
| $CH_4$ | | <0.001 | - | - |
| $H_2O$ strat. | | 0.015 | 0.51 | 0.55 |
| Feedbacks | [W m$^{-2}$ K$^{-1}$] | | | |
| $O_3$ trop. | | -0.023 | -0.029 | 0.005 |
| $O_3$ strat. | | -0.016 | -0.025 | -0.006 |
| $O_3$ total | | -0.039 | -0.054 | -0.001 |
| $CH_4$ | | -0.025 | -0.019 | 0.004 |
| $CH_4$ using PSrad | | -0.041 | -0.089 | - |
| $H_2O$ strat. | | 0.15 | 0.11 | 0.079 |

Values after the $\pm$ sign are 2$\times$ the standard error of the mean calculated on the basis of 20 annual mean values, which approximate the corresponding 95% confidence intervals. The standard errors for the climate sensitivity parameters are calculated from the standard error of the corresponding radiative forcing $std\_err_{RF}$ and the standard error of $\Delta$GSAT $std\_err_{\Delta GSAT}$, as $std\_err_\lambda = \left( \sqrt{\frac{std\_err_{RF}^2}{RF^2} + \frac{std\_err_{\Delta GSAT}^2}{\Delta GSAT^2}} \cdot \frac{\Delta GSAT}{RF} \right)$. The method to derive stratospheric adjusted radiative estimates does not account for interannual variability, which is why no uncertainty estimates are provided for the respective estimates.

model, but with prescribed $CH_4$ surface mixing ratios. In particular, to reach the 5$\times CH_4$ mixing ratio increase, a pseudo surface emission flux is calculated in their set-up. The increase factor of the pseudo flux that corresponds to an increase of 5$\times CH_4$

is 2.75 (Stecher et al., 2021), exactly the scaling factor of $CH_4$ surface emissions used in this study. To summarize, reduced



chemical decomposition enhances the increase of the $CH_4$ mixing ratios compared to the emissions. The relation between the increase of $CH_4$ mixing ratios and $CH_4$ emissions appears to be robust, if either the mixing ratio or the emissions are increased.

We separately assess the so-called fast response in $CO_2$ and $CH_4$ perturbation simulations with prescribed SSTs and SICs, and the full response in simulations coupled to a MLO model. The $CO_2$ perturbation affects the chemical composition only indirectly through temperature changes. In the fast response, radiatively induced cooling in the stratosphere causes slower chemical depletion and leads therefore to increasing mixing ratios of $O_3$ and $CH_4$. In particular, our results show that the well-known increase of upper and middle stratospheric $O_3$ (e.g., Dietmüller et al., 2014; Nowack et al., 2015; Marsh et al., 2016; Chiodo et al., 2018) is part of the fast response, and not related to the associated tropospheric warming. The $CH_4$ emission increase directly influences the chemical composition in the troposphere and stratosphere. The fast response patterns of $O_3$ and

stratospheric water vapour are consistent to the changes following the $5 \times CH_4$ mixing ratio increase (Winterstein et al., 2019) as expected from the comparable magnitude of $CH_4$ mixing ratio increase.

Despite the different effect on the chemical composition in the fast response, the isolated effect of GSAT changes induced by either the $CO_2$ or the $CH_4$ increase is consistent. Tropospheric warming shortens the atmospheric lifetime of $CH_4$. The corresponding reduction of $CH_4$ mixing ratios is explicitly simulated by the used $CH_4$ emission flux driven set-up. The explicit

reduction of $CH_4$ mixing ratios allows for secondary feedbacks of OH and $O_3$. Firstly, the $CH_4$ lifetime response implicitly includes the $CH_4$-OH feedback. Consequently, the sensitivies of the $CH_4$ lifetime per unit change of GSAT, -6.7 % $K^{-1}$ for $1.35 \times CO_2$ and -7.6 % $K^{-1}$ for $2.75 \times CH_4$, are larger in the present study compared to previous CCM results using prescribed $CH_4$ mixing ratios at the lower boundary (Voulgarakis et al., 2013; Thornhill et al., 2021a; Stecher et al., 2021). Secondly, the reduction of $CH_4$ mixing ratios results in reduced formation of $O_3$ in the troposphere. This leads to substantial differences

of the climate response of tropospheric $O_3$ between this study and previous work using prescribed $CH_4$ mixing ratios at the lower boundary (Dietmüller et al., 2014; Nowack et al., 2015; Marsh et al., 2016; Nowack et al., 2018; Chiodo et al., 2018; Stecher et al., 2021). The latter studies consistently show an increase of $O_3$ mixing ratios in the tropical upper troposphere, whereas in this study the response is either insignificantly weak or indicates a reduction of $O_3$ mixing ratios in this region. An attribution method is used to identify and quantify the processes that influence tropospheric $O_3$ under climate change:

Stronger stratosphere - troposphere exchange and larger natural emissions of $O_3$ precursors lead to increases of tropospheric $O_3$, whereas enhanced chemical loss caused by the increased tropospheric humidity and the reduction of $CH_4$ mixing ratios lead to decreases. The contribution of the individual processes depend on the representation of the process of the model. For instance, the representation of $O_3$ precursor emissions from natural sources is model dependent (e.g., Voulgarakis et al., 2013; Stevenson et al., 2020; Zanis et al., 2022). The climate response of lightning $NO_x$ emissions is uncertain so that even

the sign of the projected change depends on the used parameterization (Finney et al., 2016, 2018; Zanis et al., 2022). This has implications for the climate response of tropospheric $O_3$, for which changed lightning $NO_x$ emissions are found to be an important contribution. Most schemes used in CCMs to date project increasing lightning $NO_x$ emissions in response to tropospheric warming (Voulgarakis et al., 2013; Finney et al., 2016), which is in accordance with the results of this study, whereas a more sophisticated lightning $NO_x$ parameterization indicates a decrease of lightning $NO_x$ emissions (Finney et al.,

2018). The spatial distribution of increases of biogenic $C_5H_8$ emissions over the Amazonian region and the Congo river basin



are in qualitative agreement with the climate response of biogenic emissions simulated by other CCMs (Zanis et al., 2022, see their Fig. S6). We expect the contribution of biogenic emission changes on $O_3$ to be more important than diagnosed by the used version of the TAGGING method, as it underestimates the influence of $C_5H_8$ emissions on the diagnostic tracer $O_3$ *biogenic*. The NMHC emissions are scaled by the number of C-atoms in the molecule, i.e. 5 for $C_5H_8$, before they are added

to the NMHC family tracer, which was not done in the case of the online calculated biogenic $C_5H_8$ emissions. We expect that this issue influences the quantitative response of the category $O_3$ from biogenic sources, more precisely that changes of $O_3$ *biogenic* caused by changes of $C_5H_8$ emissions are underestimated. However, we do not expect that the response patterns and general findings would change. Therefore, and due to the computational costs of the simulations, it was decided to not repeat the simulations. We also want to stress that this issue affects the diagnostic TAGGING results only, but not the total

$O_3$ response. Furthermore, biogenic emissions of $C_5H_8$ depend on the underlying vegetation, which is expected to interact with changes in e.g. climate, atmospheric $CO_2$ abundance, tropospheric $O_3$ or land use change (e.g., Zhou et al., 2018; Vella et al., 2023), but such interactions are not included in the used set-up. In addition, droughts can inhibit the substrate supply and thereby reduce biogenic $C_5H_8$ emissions (e.g., Wang et al., 2022). A respective representation of the impact of droughts on $C_5H_8$ emissions is not included in the used model version.

We calculate the radiative effects that correspond to the composition changes of $CH_4$, $O_3$ and stratospheric $H_2O$. Our results confirm that chemical adjustments of $O_3$ and stratospheric $H_2O$ are important contributions to the ERF of the $CH_4$ perturbation. Therefore, ERF is significantly larger than SARF, which represents the direct radiative effect of $CH_4$ including the associated stratospheric temperature adjustment. The individual adjustments are in agreement with the estimates of the $5 \times CH_4$ mixing ratio experiment (Winterstein et al., 2019).

Chemical adjustments play a minor role in the $CO_2$ case. Here, stratospheric $O_3$ induces a negative adjustment, which reduces the ERF by about -1.3%. Previous studies defined the full response of $O_3$ as its feedback (Dietmüller et al., 2014; Nowack et al., 2015; Marsh et al., 2016). Our results show that under the ERF framework large parts of the stratospheric $O_3$ change is to be regarded as an adjustment. Further, our results do not indicate that ERF is significantly different from SARF for the $CO_2$ perturbation, which is in accordance to simulation results, which account for physical adjustments only (Smith et al.,

645 2018).

The scaling of the $CO_2$ and the $CH_4$ perturbations in this study are chosen so that the resulting ERFs are of similar magnitude. This allows an optimal comparison of the climate sensitivity parameters as these can depend on the magnitude of the perturbation (e.g., Hansen et al., 2005; Dietmüller et al., 2014). Our results suggest an efficacy of unity for the $CH_4$ perturbation when the climate sensitivity parameters are based on ERF. The climate sensitivity parameters based on SARF differ signifi-

cantly between the $CO_2$ and the $CH_4$ increase, because of the large effect of chemical adjustments for the $CH_4$ perturbation. Thereby, our results support the finding that the climate sensitivity is in general less dependent on the type of perturbation for the ERF framework (e.g., Richardson et al., 2019). The multi-model mean efficacy based on ERF using the fixed SST method for a $3 \times CH_4$ perturbation analyzed by Richardson et al. (2019) is slightly less than 1, with a spread of individual models of 0.56–1.15. In their study, the efficacy based on SARF is slightly less than 1 as well. In our study, the efficacy based on SARF




is significantly larger than 1, i.e. about 3.6, because of the effect of chemical adjustments, which is not included in the estimate of Richardson et al. (2019).

The feedbacks corresponding to tropospheric $O_3$ changes in the full response are negative in contrast to previous estimates derived with the EMAC model with prescribed $CH_4$ mixing ratios at the lower boundary (Dietmüller et al., 2014; Stecher et al., 2021). As mentioned above, the explicit reduction of $CH_4$ mixing ratios allows for secondary feedbacks on
$O_3$ to evolve and leads, more precisely, to reduced $O_3$ formation, which is also reflected by the negative radiative feedback. Furthermore, the feedback parameter of tropospheric $O_3$ derived from the multi-model mean analyzed by Stevenson et al. (2006) does not account for the $CH_4$ chemistry-climate feedback. The corresponding feedback parameter has been estimated at $-0.007\pm0.009$ W m$^{-2}$ K$^{-1}$ by Heinze et al. (2019).

The reduction of $CH_4$ mixing ratios in the climate response induces a negative feedback parameter, which is estimated at
$-0.025$ W m$^{-2}$ K$^{-1}$ for the $CO_2$ perturbation, and at $-0.019$ W m$^{-2}$ K$^{-1}$ for the $CH_4$ perturbation using the default radiative transfer scheme of EMAC. However, it is known that this radiative transfer scheme underestimates the direct radiative effect of $CH_4$ (Winterstein et al., 2019; Nützel et al., 2024). Therefore, we derive the radiative feedback also with the PSrad radiation scheme (Pincus and Stevens, 2013; Nützel et al., 2024), which suggests a clearly more pronounced feedback parameters of $-0.041$ W m$^{-2}$ K$^{-1}$ for the $CO_2$ perturbation, and of $-0.089$ W m$^{-2}$ K$^{-1}$ for the $CH_4$ perturbation. The PSrad feedback
parameters are in the range of previous individual model estimates derived from $CH_4$ lifetime changes (Dietmüller et al., 2014; Heinze et al., 2019; Thornhill et al., 2021a).

Further, the default radiation scheme used in EMAC so far does not account for absorption of $CH_4$ in the solar SW spectrum. Recent studies have shown that accounting for SW absorption by $CH_4$ influences adjustment and feedback processes of, e.g. clouds (Smith et al., 2018; Modak et al., 2018; Allen et al., 2023, 2024). For an improved representation of the radiative effect
of $CH_4$ in future studies, the PSrad radiation scheme (Pincus and Stevens, 2013) is now available for use in online EMAC simulations (Nützel et al., 2024).

This study focuses on the role of $CH_4$ for interactions between the gas-phase chemistry and climate change. However, further processes can also play a role, which are not accounted for by the used simulation set-up. For instance, chemistry-aerosol-cloud coupling was identified to contribute to the ERF of $CH_4$ perturbations (Kurtén et al., 2011; O'Connor et al., 2022) and might
therefore also influence the corresponding climate response. In addition, natural emission sources of $CH_4$, e.g. from wetlands or permafrost, have the potential to increase in a warming climate (e.g., O'Connor et al., 2010; Dean et al., 2018). For instance, the results of Thornhill et al. (2021a) suggest that the negative radiative effect corresponding to the shortening of the $CH_4$ lifetime is offset by the positive radiative effect of $CH_4$ emission increases from wetlands as response to a $4\times CO_2$ perturbation. The net effect of feedbacks of the gas-phase chemistry and of natural emissions influences the effect of associated secondary feedbacks,
e.g. regarding the formation of $O_3$.

To conclude, the atmospheric abundance $CH_4$, and therefore its potential as a greenhouse gas, is linked to a number of complex interactions. This makes the assessment of the climate feedback of $CH_4$ and its indirect effects a non trivial undertaking, which requires comprehensive chemistry-climate simulations. The novelty of this study is that the feedback of $CH_4$



mixing ratios to changes of its chemical sink, and thereby also associated secondary feedbacks of OH and $O_3$, are accounted

for explicitly, which is recommended to be adopted for further studies on chemical feedbacks.

*Code and data availability.* The Modular Earth Submodel System (MESSy; doi: 10.5281/zenodo.8360186) is continuously further developed and applied by a consortium of institutions. The usage of MESSy and access to the source code is licenced to all affiliates of institutions which are members of the MESSy Consortium. Institutions can become a member of the MESSy Consortium by signing the MESSy Memorandum of Understanding. More information can be found on the MESSy Consortium Website (http://www.messy-interface.org). The

simulation results presented here are based on MESSy version 2.55.2 (The MESSy Consortium, 2021). Furthermore the exact code version used to produce the simulation results is archived at the German Climate Computing Center (DKRZ) and can be made available to members of the MESSy community upon request. The simulation results are also archived at DKRZ and are available upon request. The HadISST data (Rayner et al., 2003) are available from www.metoffice.gov.uk/hadobs (last access 16 September 2024).

*Author contributions.* FW and MP developed the concept of the study. LS performed the model simulations, analyzed the data and created

the figures with significant contributions regarding the interpretation and evaluation of the model results from all coauthors. LS prepared the original draft and all authors contributed to the writing and reviewing of the manuscript.

*Competing interests.* Some authors are members of the editorial board of ACP.

*Acknowledgements.* We thank Simone Dietmüller (DLR) for doing the internal review. We used Climate Data Operators (CDO; https://code.mpimet.mpg.de/projects/cdo/, last access: 12 July 2024; Schulzweida, 2023) and netCDF Operators (NCO; Zender, 2024) for data processing.

Further, we used Python, especially the packages Xarray (Hoyer and Hamman, 2017) and Matplotlib (Hunter, 2007), for data analysis and producing the figures. We furthermore thank all contributors of the project ESCiMo (Earth System Chemistry integrated Modelling), which provides the model configuration and initial conditions. This work used resources of the Deutsches Klimarechenzentrum (DKRZ) granted by its Scientific Steering Committee (WLA) under project MIMETIC (bd1132).

*Financial support.* We acknowledge the financial support by the DFG Project IRFAM-ClimS (Vorhaben WI 5369/1-1) and the DLR internal

project MABAK (Innovative Methoden zur Analyse und Bewertung von Veränderungen der Atmosphäre und des Klimasystems).



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
