# Peer review of "Chemistry-climate feedback of atmospheric methane in a methane emission flux driven chemistry-climate model"

_EGUsphere, 2024_

## Author Comment (AC1)

**Reply to Reviewer #1**

Laura Stecher[1], Franziska Winterstein[1], Patrick Jöckel[1], Michael Ponater[1], Mariano Mertens[1], and Martin Dameris[1]

[1]Deutsches Zentrum für Luft- und Raumfahrt, Institut für Physik der Atmosphäre, Oberpfaffenhofen, Germany

**Correspondence:** Laura Stecher (laura.stecher@dlr.de)

We thank Reviewer #1 for their comments and their evaluation of our paper. Below, we repeat each comment (in blue) and address it (in black). Changes of the manuscript are written in italics.

**1 General comments**

**1.1 Ocean model coupling**

A mixed layer ocean model is coupled in the REF-SSTvar simulations. How representative are the SSTs in these runs compared to the prescribed SSTs? This would have impacts on air-sea interactions and thus affecting atmospheric chemistry and composition in addition to the GSAT. The biases in reference cases may propagate into the perturbed cases, which would affect your assessment in the climate response. Do you think the biases could be canceled out between the reference run and perturbed run?

You are right that we did not discuss possible differences between both reference simulations. The study by Stecher et al. (2021) used the same setup of the MLOCEAN submodel. As their study was one of the first to use this configuration, they provide a more detailed evaluation of differences between the reference simulations with prescribed sea surface temperatures (SSTs) and with MLO. In addition, Appendix C of the PhD thesis of the first author (Stecher, 2024) compares the reference simulation used in this study (REF-SSTfix and REF-SSTvar). We added a short text with both references in line 149. Concerning the distribution of chemical species, the zonal mean difference of methane ($CH_4$) mixing ratios between REF-SSTfix and REF-SSTvar is below 0.5% in the troposphere. Ozone ($O_3$) mixing ratios are up to 2% smaller in the southern troposphere in the MLO reference, which can be linked to a higher abundance of water vapour ($H_2O$) and thereby a stronger chemical sink. $O_3$ mixing ratios are up to 5% larger in the tropical tropopause region in the MLO reference, which is linked to a slightly larger (less than 1 hPa) tropopause pressure in the MLO simulation (see Stecher, 2024, Appendix C).

**Add in line 149**:

*Appendix C of Stecher (2024) provides a comparison of the two reference simulations. Overall, the simulation REF-SSTvar reproduces the simulation REF-SSTfix well. The largest differences are in the Southern Hemisphere (SH) polar region, where the MLO model tends to underestimate the sea ice area of the prescribed climatology, which has been noted for a similar application of the MLOCEAN submodel as well (Stecher et al., 2021).*

**1.2 Methane radiative feedback**

*The simulations between reference cases and perturbed cases suggest larger sensitivity of the methane lifetime towards climate change compared to previous studies and the authors attribute such difference to the different methane representations in the models, e.g., emission driven vs prescribed lower boundaries. But the authors also point out in the manuscript, the model-specific parameterizations, mechanisms, etc would lead to the model-dependent results. For example, using different radiative transfer models give very different radiative feedback as shown in the manuscript. How to better quantify the sensitivity to different model setup or model schemes? How to improve the model confidence in assessing methane radiative feedback?*

We think that it is important to differentiate between the sensitivity of firstly, the response of OH and $CH_4$ mixing ratios, and secondly, of the quantification of the corresponding radiative effects here. In our study, the radiative effect corresponding to the same $CH_4$ change depends strongly on the radiative transfer scheme, with the PSrad scheme providing the more reliable estimates. The response of OH and $CH_4$ could also depend on the radiation scheme driving the model, but our simulations results cannot answer this question. As we mention in the outlook, such a comparison is planned in the future.

The model spread of previously published estimates of the radiative feedback of $CH_4$ is caused by the spread of the OH response, as the $CH_4$ radiative effects are derived using formulas, e.g. by Etminan et al. (2016), (Heinze et al., 2019; Thornhill et al., 2021). The OH response is influenced by many factors and it would indeed by valuable to identify reasons for CCM differences in future studies.

In general, we think that a multi-model comparison of the $CH_4$ feedback from chemistry-climate simulations driven by $CH_4$ emission fluxes would be helpful. These estimates could be compared to previous multi-model mean estimates from simulations with $CH_4$ prescribed at the lower boundary to assess the influence of $CH_4$ emission fluxes. We modified the paragraph starting in line 601 also taking into account comment 7 and 8 by referee 2.

**Modify line 601 (see also comment 7 and 8 by referee 2):**

Modified:

*The sensitivities of the $CH_4$ lifetime per unit change of global surface air temperature (GSAT) are -6.7 % $K^{-1}$ for 1.35×carbon dioxide ($CO_2$) and -7.6 % $K^{-1}$ for 2.75×$CH_4$, which is larger compared to previous CCM results using prescribed $CH_4$ mixing ratios at the lower boundary (Voulgarakis et al., 2013; Thornhill et al., 2021; Stecher et al., 2021). The results of the comparable $CH_4$ increase experiment with prescribed $CH_4$ surface mixing ratios (Stecher et al., 2021) provides a clear indication that the lifetime change per temperature change is larger in the $CH_4$ emission driven set-up. A comparable $CO_2$ increase simulation using EMAC with prescribed $CH_4$ surface mixing ratios is not available, but the comparison to the results of other CCMs (Voulgarakis et al., 2013; Thornhill et al., 2021) indicates the same effect (see Sect. 3.1). Estimates of the $CH_4$ lifetime change per temperature change from other chemistry-climate models (CCMs) driven by prescribed $CH_4$ emission fluxes would be helpful to verify the influence of $CH_4$ emission fluxes in comparison to prescribing $CH_4$ at the lower boundary. Additionally, the multi-model differences of the $CH_4$ lifetime change per unit change of GSAT are large (Voulgarakis et al., 2013; Thornhill et al., 2021) and it would be valuable to identify reasons behind CCM differences in future studies.*

**1.3 Ozone**

 There are a lot of discussions in the manuscript on the impacts on tropospheric and stratospheric ozone. What about impacts on surface O3, which is more relevant to the health effects.

Tropospheric and stratospheric $O_3$ changes are important to understand the corresponding radiative effects and health effects are not the focus of the paper. Nevertheless, we added the response of surface ozone in the supplement and mention it shortly in the main manuscript.

**Include in line 453:**

*The spatial distribution of the climate response of surface $O_3$ is likewise similar for the $CO_2$ and the $CH_4$ perturbation (see Fig. S4).*

**2   Specific comments**

Page 9, Section 3.1, line 265-267, How sensitive of OH levels to lightning NOx? Do changes in lightning NOx play a role here?

Thank you for pointing this out. As we show later in the manuscript, lightning $NO_x$ emissions increase by about 0.3 Tg(N) a$^{-1}$ in the climate response, which is expected to lead to enhanced OH production. To estimate the effect on the $CH_4$ lifetime, we use the multi-model mean sensitivity of the $CH_4$ lifetime towards lightning $NO_x$ emission change of -4.8% (Tg(N) a$^{-1}$)$^{-1}$ from Thornhill et al. (2021), which suggest a shortening of the $CH_4$ lifetime due to the lightning $NO_x$ emission change by 1.4%. The $CH_4$ lifetime shortens by 7.4% in total. We will adapt the text as follows:

**Modify line 265:**

Previous:

*The OH response is largely driven by the increase of tropospheric humidity associated with higher temperatures.*

Adapted:

*Firstly, emissions from lightning $NO_x$ increase by about 0.3 Tg(N) a$^{-1}$ in the climate response (see Tab. 3), which leads to enhanced production of OH. To estimate the effect on the $CH_4$ lifetime, we use the multi-model mean sensitivity of the $CH_4$ lifetime towards lightning $NO_x$ emission change of -4.8% (Tg(N) a$^{-1}$)$^{-1}$ from Thornhill et al. (2021), which suggest a shortening of the $CH_4$ lifetime due to lightning $NO_x$ emissions by 1.4%. Additionally, the increase of the tropospheric humidity associated with higher temperatures leads to enhanced production of OH.*

Page 13, line 375-380: How do you treat N2O in the model? Does your model read N2O emissions or prescribe N2O at lower boundaries?

Nitrous oxide ($N_2O$) mixing ratios are prescribed at the lower boundary. We added this information to the methods section.

**Add in line 142 (see also reply to referee 2):**

*The mixing ratios of $CO_2$, $N_2O$ and ozone depleting substances (ODS) are prescribed at the lower boundary using monthly mean values of the year 2010 (Meinshausen et al., 2011; Carpenter et al., 2018). For the radiation, a CFC-11 equivalent is calculated lumping additional radiatively active ODS via radiative efficiencies following the approach by Meinshausen et al.*

*(2017). For the short-lived halocarbons* $CHCl_2Br$, $CHClBr_2$ *and* $CH_2ClBr$, *as well as* $CH_2Br_2$ *and* $CHBr_3$ *surface emissions are prescribed from Warwick et al. (2006) and Liang et al. (2010), respectively.*

95     Figure S2 shows the difference in the specific humidity. Is specific humidity in your model also affected by chemistry? In other words, is water vapor a prognostic chemical tracer explicitly involved in the chemical reactions?

    Yes, $H_2O$ is a prognostic chemical tracer in the chemistry module MECCA. Its chemical feedback modifies the prognostic specific humidity, and vice versa.

    **Add in line 126**:

[revised manuscript text omitted]

---

## Author Comment (AC2)

**Reply to Reviewer #2**

Laura Stecher[1], Franziska Winterstein[1], Patrick Jöckel[1], Michael Ponater[1], Mariano Mertens[1], and Martin Dameris[1]

[1]Deutsches Zentrum für Luft- und Raumfahrt, Institut für Physik der Atmosphäre, Oberpfaffenhofen, Germany

**Correspondence:** Laura Stecher (laura.stecher@dlr.de)

We thank Reviewer #2 for their comments and their evaluation of our paper. Below, we repeat each comment (in blue) and address it (in black). Changes of the manuscript are written in italics.

**1 Specific comments**

1. In Section 2.1, you state that chlorine and bromine halogen chemistry is included and that oxidation of methane by chlorine is considered. Can you include some clarification on whether the chlorine sink is only relevant in the stratosphere and/or whether the methane sink in the marine boundary layer through tropospheric halogen chemistry is included?

   The methane sink through the reaction with chlorine is accounted for in the whole atmosphere. However, the reaction with chlorine accounts for only about 0.23% of the total tropospheric methane ($CH_4$) loss in our reference simulations (see Table S1, which we added to the supplement). We added this information in the Methods section. We also added the following information about emissions of the short-lived halogens $CHCl_2Br$, $CHClBr_2$, $CH_2ClBr$, $CH_2Br_2$ and $CHBr_3$, as these are expected to affect the chlorine sink.

   – **Add in line 123:**

     *Oxidation by the hydroxyl radical (OH) dominates the tropospheric $CH_4$ sink, so that the reaction with chlorine (Cl) accounts for only 0.23% of the total chemical tropospheric $CH_4$ loss (see Tab. S1 in the supplement). Therefore, we focus on $CH_4$ lifetime changes with respect to oxidation by OH.*

   – **Add in line 142 (see also reply to referee 1):**

     *The mixing ratios of carbon dioxide ($CO_2$), nitrous oxide ($N_2O$) and ozone depleting substances (ODS) are prescribed at the lower boundary using monthly mean values of the year 2010 (Meinshausen et al., 2011; Carpenter et al., 2018). For the radiation, a CFC-11 equivalent is calculated lumping additional radiatively active ODS via radiative efficiencies following the approach by Meinshausen et al. (2017). For the short-lived halocarbons $CHCl_2Br$, $CHClBr_2$ and $CH_2ClBr$, as well as $CH_2Br_2$ and $CHBr_3$ surface emissions are prescribed from Warwick et al. (2006) and Liang et al. (2010), respectively.*

2. In Section 2.1, what criteria were used to determine whether "a quasi-equilibrium is reached"?

We added the following information.

**Add in line 168 (see also community comment by Zosia Staniaszek) :**

*Time series of the global mean surface $CH_4$, the total atmospheric masses of $CH_4$ and ozone ($O_3$), the TOA radiation balance, and GSAT (for the MLO simulations) were monitored to decide whether an equilibrium is reached. In addition, we assessed the spin-up of the mass of $CH_4$ of the simulation $ERFCH_4$ in more detail. A curve fit was applied to the spin-up period to derive the atmospheric mass of $CH_4$ in equilibrium. The mass of $CH_4$ follows the exponential function of the form $a - b \cdot exp(-t/c)$ closely. The mass of $CH_4$ in the last year of the spin-up, simulation year 90, is about 0.5% smaller than the derived equilibrium estimate (parameter a), and therefore spun-up sufficiently well (see Fig. S14). The derived perturbation lifetime (parameter c) is 21.6 years. We note that the perturbation lifetime is larger than that of the $CH_4$ emission reduction experiment by Staniaszek et al. (2022). As the perturbation lifetime increases with increasing $CH_4$ burden (Holmes, 2018), this can be expected. In addition, model differences and the magnitude of the emission change might play a role.*

3. Line 240: You state that the radiative effects of ozone and water vapour are calculated separately for the troposphere and stratosphere – can you include what definition you use for the tropopause?

The climatological tropopause ($tp_{clim} = 300$ hPa $- 215$ hPa $\cdot \cos^2(\phi)$) is used. We added the missing information.

**Add in line 240:**

*To define the region in which the stratospheric temperature adjustment is applied, as well as to separate tropospheric and stratospheric radiative effects of $O_3$ and the specific humidity, the climatological tropopause $tp_{clim}$ is used consistently with the $CH_4$ lifetime calculation (see Sect 2.1).*

4. Section 3.1, line 262: You mention that the lifetime with respect to OH oxidation is reduced when the model is allowed to respond to the CO2 perturbation. Although stratospheric oxidation is a more minor sink for methane than tropospheric oxidation by OH, I wondered whether you could also diagnose the stratospheric lifetime

We calculated the stratospheric $CH_4$ lifetime using Eq. 2, analogously to the tropospheric lifetime, but taking into account all grid boxes above the tropopause. The results of the $CH_4$ loss, i.e. $\frac{1}{\tau}$, are shown in Tab. S1 in the supplement.

We mentioned the following points in the manuscript regarding the stratospheric $CH_4$ loss:

**Add in line 256:**

*The stratospheric $CH_4$ loss by reaction with OH, Cl, and excited oxygen ($O(^1D)$) is reduced by about 2% (see Tab. S1 in the supplement).*

**Add in line 267:**

*... and the stratospheric $CH_4$ loss does not change (see Tab. S1 in the supplement), ...*

**Add in line 400:**

*In the stratosphere, $CH_4$ loss by OH is enhanced due to an increase in stratospheric $HO_x$, whereas $CH_4$ loss by Cl is*

*reduced (see Tab. S1 in the supplement). Overall, chemical stratospheric $CH_4$ loss is increased by 17.5%. However, the increase of $CH_4$ mixing ratios is larger than the increase factor of surface emissions of 2.75 in the whole stratosphere as well.*

5. Section 3.1, line 269: Given the lack of significant differences in the oxidants, you hypothesize that the change in methane concentrations in the lower stratosphere are due to transport, i.e., because of reduced tropospheric concentrations and a more efficient Brewer Dobson circulation. Do you have any mass flux diagnostics that can support that statement?

We added a Figure showing the response of the residual streamfunction to the supplement and referred to it in the text as follows. Just to be clear, we think that enhanced tropical upwelling alone would lead to an increase of the $CH_4$ mixing ratio in the tropical lower stratosphere as air masses with larger $CH_4$ mixing ratio are transported from the troposphere into the stratosphere more efficiently.

**Add in line 270:** *Tropical upwelling is enhanced in the climate response (see Fig. S13 in the supplement).*

**Add in line 447::** *(i.e. strengthening of the Brewer-Dobson circulation, see Fig. S13 in the supplement)*

6. Section 3.1: It is interesting that the sensitivity of the methane lifetime to climate change in EMAC appears to be stronger than in Voulgarakis et al. and Thornhill et al. I'm not convinced that it is due to methane being more fully interactive here, and it would make an interesting follow-up study to try to unpick the reasons behind these model differences.

We agree that it would be valuable to identify reasons for the multi-model differences in future studies. We added this in the outlook, see our answer to comment 7.

7. Section 3.1, lines 303-305: You state that ", the explicit treatment of the CH4 feedback in our set-up allows for a subsequent feedback of OH and correspondingly for a self-feedback on the CH4 lifetime, which can explain the enhanced sensitivity of the CH4 lifetime towards climate change." Have you verified in this setup that if methane was driven by concentration-based boundary conditions that the sensitivity of methane lifetime to temperature would be more comparable to that in other models?

We agree that the comparison with a $CO_2$ increase experiment with EMAC with concentration-based boundary for $CH_4$ would be desirable. Unfortunately, we do not know of a suitable simulation for which the $CH_4$ lifetime change per temperature change was calculated. For the experiments of Dietmüller et al. (2014) the $CH_4$ lifetime change is not available anymore.

However, for the $CH_4$ increase experiment, the results of Stecher et al. (2021) provide a fair comparison with concentration-based boundary for $CH_4$. There is a clear indication that the lifetime change per temperature change is larger in the $CH_4$ emission driven set-up. We modified the statement in line 601.

**Modify line 601 (see also comment 7 and comment 1.2 by referee 1):**
Previous:
*Consequently, the sensitivities of the $CH_4$ lifetime per unit change of global surface air temperature (GSAT), -6.7 % $K^{-1}$*

90 *for 1.35×$CO_2$ and -7.6 % $K^{-1}$ for 2.75×$CH_4$, are larger in the present study compared to previous CCM results using prescribed $CH_4$ mixing ratios at the lower boundary (Voulgarakis et al., 2013; Thornhill et al., 2021a; Stecher et al., 2021).*

Modified:

*The sensitivities of the $CH_4$ lifetime per unit change of GSAT are -6.7 % $K^{-1}$ for 1.35×$CO_2$ and -7.6 % $K^{-1}$ for*
95 *2.75×$CH_4$, which is larger compared to previous CCM results using prescribed $CH_4$ mixing ratios at the lower boundary (Voulgarakis et al., 2013; Thornhill et al., 2021a; Stecher et al., 2021). The results of the comparable $CH_4$ increase experiment with prescribed $CH_4$ surface mixing ratios (Stecher et al., 2021) provides a clear indication that the lifetime change per temperature change is larger in the $CH_4$ emission driven set-up. A comparable $CO_2$ increase simulation using EMAC with prescribed $CH_4$ surface mixing ratios is not available, but the comparison to the results of other CCMs (Voulgarakis et al., 2013; Thornhill et al., 2021a) indicates the same effect (see Sect. 3.1). Estimates of the*
100 *$CH_4$ lifetime change per temperature change from other chemistry-climate models (CCMs) driven by prescribed $CH_4$ emission fluxes would be helpful to verify the influence of $CH_4$ emission fluxes in comparison to prescribing $CH_4$ at the lower boundary. Additionally, the multi-model differences of the $CH_4$ lifetime change per unit change of GSAT are large (Voulgarakis et al., 2013; Thornhill et al., 2021a) and it would be valuable to identify reasons behind CCM differences in future studies.*

105 8. Section 3.1, lines 306-310: Here, you argue that the model response in EMAC is more consistent with f=1 than estimates of f = [1.2, 1.4]. I wonder how representative the range of 1.2-1.4 is for the EMAC model. Do you know what the feedback factor from EMAC is in concentration-driven simulations or even from your ch4 flux perturbation simulation?

We calculated the feedback factor $f$ from the present $CH_4$ emission increase simulation, which suggests $f$=1.55 (see also reply to community comment by Zosia Staniaszek). This is at the larger end of previously published estimates. The
110 feedback factor $f$ increases with increasing $CH_4$ burden (Holmes, 2018), so that $f$=1.55 is most likely not representative for smaller $CH_4$ perturbations with the EMAC model. Applying $f$=1.55 implies an even larger reduction of $CH_4$ mixing ratio to 1.61 ppmv.

We applied the formula here to show that for the $CH_4$ emission driven simulation, the $CH_4$-OH feedback is already included in the OH and the $CH_4$ lifetime responses. Therefore, using the formula with $f$>1 applies the $CH_4$-OH feedback
115 twice, which results in a larger $CH_4$ reduction than simulated by the model. We rephrased the paragraph and extended the range of $f$.

**Modify in line 306:**

Previous:

*If the $CH_4$ mixing ratio can not adapt to changes in its lifetime, the corresponding $CH_4$ equilibrium mixing ratio can be*
120 *estimated using Eq. 1, which indicates a global mean $CH_4$ equilibrium mixing ratio in the range of 1.63 to 1.66 parts per million volume (ppmv) for f = [1.2, 1.4] for the present changes of the $CH_4$ lifetime. Thus, Eq. 1 suggests a larger reduction than simulated by the model, which adjusts to a global mean $CH_4$ equilibrium mixing ratio of 1.69 ppmv*

*(see Tab. 2). However, if the feedback factor is not applied (f=1), Eq. 1 gives 1.68 ppmv, which is in close agreement with the simulated response of $CH_4$ mixing ratios and supports the assumption that the sensitivity of OH and the $CH_4$ lifetime towards climate change is larger, if the feedback of $CH_4$ is explicitly simulated as thereby the $CH_4$-OH feedback is implicitly included in the simulated response.*

Modified:

*If the $CH_4$ mixing ratio can not adapt to changes in its lifetime, the corresponding $CH_4$ equilibrium mixing ratio can be estimated using Eq. 1. The feedback factor f in the equation accounts for the $CH_4$-OH feedback. In our $CH_4$ emission driven simulation the $CH_4$-OH feedback is implicitly included in the simulated response of OH and the $CH_4$ lifetime, so that using Eq. 1 with f > 1 applies the $CH_4$-OH feedback twice in this case. Eq. 1 indicates a global mean $CH_4$ equilibrium mixing ratio in the range of 1.61 to 1.66 ppmv for f = [1.19, 1.55] for the present changes of the $CH_4$ lifetime. Thus, it suggests a larger reduction than simulated by the model, which adjusts to a global mean $CH_4$ equilibrium mixing ratio of 1.69 ppmv (see Tab. 2). However, if the feedback factor is not applied (f=1), Eq. 1 gives 1.68 ppmv, which is in close agreement with the simulated response of $CH_4$ mixing ratios.*

9. Section 3.3: Here you state that Table 4 shows the total SARF, ERF, $\Delta$GSAT and the associated climate sensitivity parameters $\lambda$, as well as individual radiative effects corresponding to the composition changes of CH4, O3 and stratospheric H2O. You then go on to compare SARF and ERF for the co2 and ch4 perturbation simulations. In the case of the CH4 perturbation simulation, the ERF is a factor of 3 larger than the SARF. I think it's important to state upfront that the SARF here is only capturing the direct radiative effect of CH4 alone, whereas the ERF captures the radiative effects from the ch4-driven chemical adjustments (e.g., ozone, stratospheric water vapour).

We added a clarification at the beginning of Sect. 3.3. and in the caption of Table 4.

**Add in line 487 and in caption of Table 4:**

*ERF includes physical and chemical adjustments, whereas SARF represents the radiative effect of the $CO_2$ or $CH_4$ composition change and the corresponding stratospheric temperature adjustment only.*

10. Section 3.3: Here, the radiative effect of stratospheric water vapour from the methane flux perturbation experiment seems to be nearly comparable in magnitude with that from ozone. This doesn't appear to be consistent with the relative contributions from water vapour and ozone in the present-day forcing from methane. Can you comment further on its radiative effect?

We assume that you are referring to the studies by Thornhill et al. (2021b); O'Connor et al. (2022), which quantify the contribution of $O_3$ and SWV to the pre-industrial to present-day $CH_4$ radiative forcing. The relative contribution of $O_3$ is about 16-27% and 13-21%, and of SWV about 0.5% and 2-7% in the studies by Thornhill et al. (2021b) and O'Connor et al. (2022), respectively. So the ratio $\frac{RF_{SWV}}{RF_{O3}}$ is 0.02 for Thornhill et al. (2021b), and 0.10 (Table 3, Single forcing method) and 0.54 (Table 2, Elimination method) for O'Connor et al. (2022). In our study, the relative contribution of $O_3$ is about 47% and of SWV 30%. For both, $O_3$ and SWV, the relative contributions are overestimated as the direct $CH_4$ radiative forcing is underestimated, which we discuss in the manuscript in detail. $\frac{RF_{SWV}}{RF_{O3}} = 0.63$ in our study, which

is larger than in previous studies, but somewhat comparable in magnitude to the results of the Elimination method by O'Connor et al. (2022).

In general, it is difficult to decide whether model/methodological differences or the different time period/magnitude of the perturbation causes the difference. In addition, the differences in the radiative forcing could be caused by the sensitivity of the water vapour response, or by the sensitivity of the water vapour radiative forcing. For example, it makes a difference if the associated stratospheric temperature adjustment is included. We are unsure if the corresponding stratospheric temperature adjustment is included in the estimate provided by Thornhill et al. (2021b). Thus, we decided to not discuss SWV radiative forcing differences in the text as model, methodological and forcing differences can play a role here. A more targeted multi-model/multi-method initiative could help to attribute and to better understand differences.

**2 Technical comments**

Thank you for the suggested improvements for wording. We have adopted the suggestions. For the following comments, we deviated slightly from the suggestion.

- Comment 22: We reformulated the paragraph and hope that it is easier to understand now.

- Comment 36: The sentence reads now. *Biogenic emissions of* $NO_x$ *and* $C_5H_8$ *increase in the full response as well*.

- Comment 48: We changed the occurrence in line 688, but not in line 687 as in line 687 the expression *climate feedback of CH₄* refers to the change in $CH_4$ mixing ratios (*feedback on CH₄ mixing ratios*), but also to the corresponding radiative feedback (*of CH₄*).

**References**

[revised manuscript text omitted]

---

## Author Comment (AC3)

**Reply to Zosia Staniaszek**

Laura Stecher[1], Franziska Winterstein[1], Patrick Jöckel[1], Michael Ponater[1], Mariano Mertens[1], and Martin Dameris[1]

[1]Deutsches Zentrum für Luft- und Raumfahrt, Institut für Physik der Atmosphäre, Oberpfaffenhofen, Germany

**Correspondence:** Laura Stecher (laura.stecher@dlr.de)

We thank Zosia Staniaszek for her community comment. Below, we repeat each comment (in blue) and address it (in black). Changes of the manuscript are written in italics.

Thanks for a really interesting read and an important step in developing more models with methane emissions. The section

5  on tagging the ozone response is particularly novel in a methane emissions driven system and really illustrates the wide ranging impact of changes in methane (that are often excluded by using a lower boundary condition).

Thank you for the positive assessment of our study.

The increase in mixing ratio above the level of increase in emissions is as expected due to the chemical feedbacks in the

10  system and the lengthening of methane lifetime, albeit here the feedback factor is higher than most models (1.73 as a crude estimate using delta conc/delta emissions (Holmes et al 2018)). I would be interested in whether you could calculate a feedback factor for EMAC using these simulations. Also, a note that more recent feedback factors than those quoted here can be found in Sand et al 2023 Supplementary Table 2 (https://www.nature.com/articles/s43247-023-00857-8).

15  Thank you for pointing us to the study of Sand et al., 2023. We added the reference to the introduction. The respective sentence in line 56 reads now: *Estimates of f are in the range of 1.19 to 1.55 (Fiore et al., 2009; Voulgarakis et al., 2013; Stevenson et al., 2013; Thornhill et al., 2021b; Stevenson et al., 2020; Sand et al., 2023)*

We calculated the feedback factor using Eq. 12 of Holmes (2018), and also using a curve fit of the spin-up of $ERFCH_4$ (see below). The feedback factor of 1.55 is most likely not representative for smaller methane ($CH_4$) perturbations or smaller $CH_4$

20  burden, which might explain why it is at the larger end of previous estimates. We added the following to Sect. 3.2..

**Add in line 399:**

*We derive the feedback factor $f$ (see Eq. 1) from $ERFCH_4$ using two approaches. Firstly, it is calculated from Eq. 12 by Holmes (2018) as $f = \frac{ln(m_1/m_0)}{ln(E_1/E_0)}$. Secondly, it is derived from a curve fit of the function $m(t) = m_0 * [2.75^f + (2.75 - 2.75^f) *$*

25  *$exp(-t/(f*\tau)]$ (Holmes, 2018) of the spin-up of the atmospheric mass of $CH_4$ using the yearly mean $CH_4$ lifetime with respect to OH oxidation for $\tau$ (see Fig. S14). Both approaches suggest $f = 1.55$. However, the derived $f$ is not expected to be representative for $CH_4$ perturbations of EMAC close to present-day conditions because $f$ increases with increasing $CH_4$*

*burden (Holmes, 2018). This might also explain why our estimate of $f$ is at the upper end of previous estimates (Fiore et al., 2009; Voulgarakis et al., 2013; Stevenson et al.,2013; Thornhill et al., 2021b; Stevenson et al., 2020; Sand et al., 2023).*

30

I would also be interested in the timescale of the change in methane mixing ratio, and the perturbation lifetime. In the methods section you mention a long spin up time. In UKESM-ems we found that with a large methane emissions decrease you get a much faster than expected change in mixing ratio due to the rapid increase in OH and decrease in methane lifetime, and I would expect the opposite effect in an increase such as done here.

35

We added the following information about the spin-up of the simulation ERFCH$_4$. In particular, the spin-up of the mass of CH$_4$ follows the exponential function of the form $a - b \cdot exp(-t/c)$ closely. The fitted perturbation lifetime (parameter $c$) is 21.63 years. After 90 years the mass of CH$_4$ is 0.5% smaller than the expected equilibrium mass (parameter $a$). We even shortened the spin-up by initializing the atmospheric CH$_4$ mixing ratios with the reference values scaled by a factor of 2.75

40 (the increase factor of the emissions).

We additionally fitted $f$ using the function $m(t) = m_0 * [2.75^f + (2.75 - 2.75^f) * exp(-t/(f * \tau))]$ (Holmes, 2018) using the yearly mean CH$_4$ lifetime with respect to OH oxidation for $\tau$, which suggest f=1.55. The perturbation lifetime corresponding to the last 5 years of the spin-up is 22.4 years. Both approaches do not account for the dependence of $f$ on the CH$_4$ burden (Holmes, 2018), as a constant $f$ is fitted.

45 The perturbation lifetime corresponding to our emission increase simulation is larger than that of the emission reduction by Staniaszek et al. (2022), to which we think you are referring. We think that it is plausible that the perturbation lifetime is shorter for a CH$_4$ emission reduction as $f$ and $\tau$ depend on the CH$_4$ burden Holmes (2018). Therefore, also the perturbation lifetime $f \cdot \tau$ depends on the CH$_4$ burden. In our emission increase simulation, the increase of the CH$_4$ lifetime is large, and therefore also the extension of the perturbation lifetime is large. It would be interesting to investigate this in targeted simulations, e.g.,

50 branching of both, a CH$_4$ emission reduction and emission increase of the same magnitude, from an identical reference.

**Add in line 168 (see also comment 2 by referee 2):**

*Time series of the global mean surface CH$_4$, the total atmospheric masses of CH$_4$ and ozone (O$_3$), the TOA radiation balance, and GSAT (for the MLO simulations) were monitored to decide whether an equilibrium is reached. In addition, we assessed the*

55 *spin-up of the mass of CH$_4$ of the simulation ERFCH$_4$ in more detail. A curve fit was applied to the spin-up period to derive the atmospheric mass of CH$_4$ in equilibrium. The mass of CH$_4$ follows the exponential function of the form $a - b \cdot exp(-t/c)$ closely. The mass of CH$_4$ in the last year of the spin-up, simulation year 90, is about 0.5% smaller than the derived equilibrium estimate (parameter a), and therefore spun-up sufficiently well (see Fig. S14). The derived perturbation lifetime (parameter c) is 21.6 years. We note that the perturbation lifetime is larger than that of the CH$_4$ emission reduction experiment by Staniaszek*

60 *et al. (2022). As the perturbation lifetime increases with increasing CH$_4$ burden (Holmes, 2018), this can be expected. In addition, model differences and the magnitude of the emission change might play a role.*

**References**

Holmes, C. D.: Methane Feedback on Atmospheric Chemistry: Methods, Models, and Mechanisms, J. Adv. Model. Earth Syst., 10, 1087–1099, https://doi.org/10.1002/2017MS001196, 2018.

65  Staniaszek, Z., Griffiths, P. T., Folberth, G. A., O'Connor, F. M., Abraham, N. L., and Archibald, A. T.: The role of future anthropogenic methane emissions in air quality and climate, npj Clim Atmos Sci, 5, https://doi.org/10.1038/s41612-022-00247-5, 2022.